# Online Sinkhorn: Optimal Transport distances from sample streams

**Arthur Mensch**
PSL University
CNRS, ENS, DMA
Paris, France
`arthur.mensch@m4x.org`

**Gabriel Peyré**
PSL University
CNRS, ENS, DMA
Paris, France
`gabriel.peyre@ens.fr`

## Abstract

Optimal Transport (OT) distances are now routinely used as loss functions in ML tasks. Yet, computing OT distances between arbitrary (i.e. not necessarily discrete) probability distributions remains an open problem. This paper introduces a new online estimator of entropy-regularized OT distances between two such arbitrary distributions. It uses streams of samples from both distributions to iteratively enrich a non-parametric representation of the transportation plan. Compared to the classic Sinkhorn algorithm, our method leverages new samples at each iteration, which enables a consistent estimation of the true regularized OT distance. We provide a theoretical analysis of the convergence of the online Sinkhorn algorithm, showing a nearly-$\mathcal{O}(\frac{1}{n})$ asymptotic sample complexity for the iterate sequence. We validate our method on synthetic 1-d to 10-d data and on real 3-d shape data.

Optimal transport (OT) distances are fundamental in statistical learning, both as a tool for analyzing the convergence of various algorithms (Canas and Rosasco, 2012; Dalalyan and Karagulyan, 2019), and as a data-dependent term for tasks as diverse as supervised learning (Frogner et al., 2015), unsupervised generative modeling (Arjovsky et al., 2017) or domain adaptation (Courty et al., 2016). OT lifts a distance over data points living in a space $\mathcal{X}$ into a distance on the space $\mathcal{P}(\mathcal{X})$ of probability distributions over the space $\mathcal{X}$. This distance has many favorable geometrical properties. In particular it allows one to compare distributions having disjoint supports. Computing OT distances is usually performed by sampling once from the input distributions and solving a discrete linear program (LP), due to Kantorovich (1942). This approach is numerically costly and statistically inefficient (Weed and Bach, 2019). Furthermore, the optimisation problem depends on a fixed sampling of points from the data. It is therefore not adapted to machine learning settings where data is resampled continuously (e.g. in GANs), or accessed in an online manner. In this paper, we develop an efficient online method able to estimate OT distances between continuous distributions. It uses a stream of data to refine an approximate OT solution, adapting the regularized OT approach to an online setting.

To alleviate both the computational and statistical burdens of OT, it is common to regularize the Kantorovich LP. The most successful approach in this direction is to use an entropic barrier penalty. When dealing with discrete distributions, this yields a problem that can be solved numerically using Sinkhorn-Knopp's matrix balancing algorithm (Sinkhorn, 1964; Sinkhorn and Knopp, 1967). This approach was pushed forward for ML applications by Cuturi (2013). Sinkhorn distances are smooth and amenable to GPU computations, which make them suitable as a loss function in model training (Frogner et al., 2015; Mensch et al., 2019). The Sinkhorn algorithm operates in two distinct phases: draw samples from the distributions and evaluate a pairwise distance matrix in the first phase; balance this matrix using Sinkhorn-Knopp iterations in the second phase.

This two-step approach does not estimate the true regularized OT distance, and cannot handle samples provided as a stream, e.g. renewed at each training iteration of an outer algorithm. A cheap fix is to use Sinkhorn over mini-batches (see for instance Genevay, Peyré, et al. (2018) for an application

to generative modelling). Yet this introduces a strong estimation bias, especially in high dimension —see Fatras et al. (2019) for a mathematical analysis. In contrast, we use streams of mini-batches to progressively enrich a consistent representation of the transport plan.

**Contributions.** Our paper proposes a new take on estimating optimal transport distances between continuous distributions. We make the following contributions:

- We introduce an online variant of the Sinkhorn algorithm, that relies on streams of samples to enrich a non-parametric functional representation of the dual regularized OT solution.
- We establish the almost sure convergence of online Sinkhorn and derive asymptotic convergence rates (Proposition 3 and 4). We provide convergence results for variants.
- We demonstrate the performance of online Sinkhorn for estimating OT distances between continuous distributions and for accelerating the early phase of discrete Sinkhorn iterations.

**Notations.** We denote $\mathcal{C}(\mathcal{X})$ $[\mathcal{C}_+(\mathcal{X})]$ the set of [strictly positive] continuous functions over a metric space $\mathcal{X}$, $\mathcal{M}^+(\mathcal{X})$ and $\mathcal{P}(\mathcal{X})$ the set of positive and probability measures on $\mathcal{X}$, respectively.

## 1 Related work

**Sinkhorn properties.** The Sinkhorn algorithm computes $\varepsilon$-accurate approximations of OT in $O(n^2/\varepsilon^3)$ operations for $n$ samples (Altschuler et al., 2017) (in contrast with the $\mathcal{O}(n^3)$ complexity of exact OT Goldberg and Tarjan, 1989). Moreover, Sinkhorn distances suffer less from the curse of dimensionality (Genevay, Chizat, et al., 2019), since the average error using $n$ samples decays like $\mathcal{O}(\varepsilon^{-d/2}/\sqrt{n})$ in dimension $d$, in contrast with the slow $\mathcal{O}(1/n^{1/d})$ error decay of OT (Dudley, 1969; Weed and Bach, 2019). Sinkhorn distances can further be sharpened by entropic debiasing (Feydy et al., 2019). Our work is orthogonal, as we focus on estimating distances between continuous distributions.

**Continuous optimal transport.** Extending OT computations to arbitrary distributions (possibly having continuous densities) without relying on a fixed a priori sampling is an emerging topic of interest. A special case is the semi-discrete setting, where one of the two distributions is discrete. Without regularization, over an Euclidean space, this can be solved efficiently using the computation of Voronoi-like diagrams (Mérigot, 2011). This idea can be extended to entropic-regularized OT (Cuturi and Peyré, 2018), and can also be coupled with stochastic optimization methods (Genevay, Cuturi, et al., 2016) to tackle high=dimensional problems (see Staib et al., 2017 for an extension to Wasserstein barycenters). When dealing with arbitrary continuous densities, that are accessed through a stream of random samples, the challenge is to approximate the (continuous) dual variables of the regularized Kantorovich LP using parametric or non-parametric classes of functions. For application to generative model fitting, one can use deep networks, which leads to an alternative formulation of Generative Adversarial Networks (GANs) (Arjovsky et al., 2017) (see also Seguy et al. (2018) for an extension to the estimation of transportation maps). There is however no theoretical guarantees for this type of dual approximations, due to the non-convexity of the resulting optimization problem. To our knowledge, the only mathematically rigorous algorithm represents potentials in reproducing Hilbert space (Genevay, Cuturi, et al., 2016). This approach is generic and does not leverage the specific structure of the OT problem, so that in practice its convergence is slow. We show in Section §5.1 that online Sinkhorn finds better potential estimates than SGD on RKHS representations.

**Stochastic approximation (SA).** Our approach may be seen as SA (Robbins and Monro, 1951) for finding the roots of an operator in a non-Hilbertian functional space. Alber et al., 2012 studies SA for solving fixed-points that are contractant in Hilbert spaces. Online Sinkhorn convergence relies on the contractivity of a certain operator in a non-Hilbertian metric, and requires a specific analysis. As both are SA instances, the online Sinkhorn algorithm resembles stochastic EM (Celeux and Diebolt, 1992), though it cannot be interpreted as such.

## 2 Background: optimal transport distances

We first recall the definition of optimal transport distances between arbitrary distributions (i.e. not necessarily discrete), then review how these are estimated using a finite number of samples.

## 2.1 Optimal transport distances and algorithms

**Wasserstein distances.** We consider a complete metric space $(\mathcal{X}, d)$ (assumed to be compact for simplicity), equipped with a continuous cost function $(x, y) \in \mathcal{X}^2 \to C(x, y) \in \mathbb{R}$ for any $(x, y) \in \mathcal{X}^2$ (assumed to be symmetric also for simplicity). Optimal transport lifts this *ground cost* into a cost between probability distributions over the space $\mathcal{X}$. The Wasserstein cost between two probability distributions $(\alpha, \beta) \in \mathcal{P}(\mathcal{X})^2$ is defined as the minimal cost required to move each element of mass of $\alpha$ to each element of mass of $\beta$. It rewrites as the solution of a linear problem (LP) over the set of transportation plans (which are probability distribution $\pi$ over $\mathcal{X} \times \mathcal{X}$):

$$\mathcal{W}_{C,0}(\alpha, \beta) \triangleq \min_{\pi \in \mathcal{P}(\mathcal{X}^2)} \{ \langle C, \pi \rangle \ : \ \pi_1 = \alpha, \pi_2 = \beta \},$$

where we denote $\langle C, \pi \rangle \triangleq \int C(x, y) \mathrm{d}\pi(x, y)$. Here, $\pi_1 = \int_{y \in \mathcal{X}} \mathrm{d}\pi(\cdot, y)$ and $\pi_2 = \int_{x \in \mathcal{X}} \mathrm{d}\pi(x, \cdot)$ are the first and second marginals of the transportation plan $\pi$. We refer to Santambrogio, 2015 for a review on OT.

**Entropic regularization and Sinkhorn algorithm.** The solutions of (1) can be approximated by a strictly convex optimisation problem, where an entropic term is added to the linear objective to force strict convexity. The so-called Sinkhorn cost is then

$$\mathcal{W}_{C,\varepsilon}(\alpha, \beta) \triangleq \min_{\substack{\pi \in \mathcal{P}(\mathcal{X} \times \mathcal{X}) \\ \pi_1 = \alpha, \pi_2 = \beta}} \langle C, \pi \rangle + \varepsilon \mathrm{KL}(\pi | \alpha \otimes \beta), \tag{1}$$

where the Kulback-Leibler divergence is defined as $\mathrm{KL}(\pi | \alpha \otimes \beta) \triangleq \int \log(\frac{\mathrm{d}\pi}{\mathrm{d}\alpha \mathrm{d}\beta}) \mathrm{d}\pi$ (which is thus equal to the mutual information of $\pi$). $\mathcal{W}_{C,\varepsilon}$ approximates $\mathcal{W}_{C,0}(\alpha, \beta)$ up to an $\varepsilon \log(\varepsilon)$ error (Genevay, Chizat, et al., 2019). In the following, we set $\varepsilon$ to 1 without loss of generality, as $\mathcal{W}_{C,\varepsilon} = \varepsilon \mathcal{W}_{C/\varepsilon, 1}$, and simply write $\mathcal{W}$. (1) admits a dual form, which is a maximization problem over the space of continuous functions:

$$F_{\alpha,\beta}(f, g) \triangleq \max_{(f,g) \in \mathcal{C}(\mathcal{X})^2} \langle f, \alpha \rangle + \langle g, \beta \rangle - \langle e^{f \oplus g - C}, \alpha \otimes \beta \rangle + 1, \tag{2}$$

where $\langle f, \alpha \rangle \triangleq \int f(x) \mathrm{d}\alpha(x)$ and $(f \oplus g - C)(x, y) \triangleq f(x) + g(y) - C(x, y)$. Problem (2) can be solved by closed-form alternated maximization, which corresponds to Sinkhorn's algorithm. At iteration $t$, the updates are simply

$$f_{t+1}(\cdot) = T_\beta(g_t), \quad g_{t+1}(\cdot) = T_\alpha(f_{t+1}),$$

$$T_\mu(h) \triangleq -\log \int_{y \in \mathcal{X}} \exp(h(y) - C(\cdot, y)) \mathrm{d}\mu(y). \tag{3}$$

The operation $h \mapsto T_\mu(h)$ maps a continuous function to another continuous function, and is a smooth approximation of the celebrated $C$-transform of OT (Santambrogio, 2015). We thus refer to it as a *soft $C$-transform*. Note that we consider *simultaneous* updates of $f_t$ and $g_t$ in this paper, as it simplifies our analysis. The notation $f_t(\cdot)$ emphasizes the fact that $f_t$ and $g_t$ are *functions*.

It can be shown that $(f_t)_t$ and $(g_t)_t$ converge in $(\mathcal{C}(\mathcal{X}), \|\cdot\|_{\mathrm{var}})$ to a solution $(f^\star, g^\star)$ of (2), where $\|f\|_{\mathrm{var}} \triangleq \max_x f(x) - \min_x f(x)$ is the so-called variation norm. Functions endowed with this norm are only considered up to an additive constant. Global convergence is due to the strict contraction of the operators $T_\beta(\cdot)$ and $T_\alpha(\cdot)$ in the space $(\mathcal{C}(\mathcal{X}), \|\cdot\|_{\mathrm{var}})$ (Lemmens and Nussbaum, 2012).

## 2.2 Estimating OT distances with realizations

When the input distributions are discrete (or equivalently when $\mathcal{X}$ is a finite set), i.e. $\alpha = \frac{1}{n} \sum_{i=1}^n \delta_{x_i}$ and $\beta = \frac{1}{n} \sum_{i=1}^n \delta_{y_i}$, the function $f_t$ and $g_t$ need only to be evaluated on $(x_i)_t$ and $(y_i)_i$, which allows a proper implementation. The iterations (3) then correspond to the Sinkhorn and Knopp (1967) algorithm over the inverse scaling vectors $\boldsymbol{u}_t \triangleq (e^{-f_t(x_i)})_{i=1}^n, \boldsymbol{v}_t \triangleq (e^{-g_t(y_i)})_{i=1}^n$:

$$\boldsymbol{u}_{t+1} = \boldsymbol{K} \frac{1}{n\boldsymbol{v}_t} \quad \text{and} \quad \boldsymbol{v}_{t+1} = \boldsymbol{K}^\top \frac{1}{n\boldsymbol{u}_t} \tag{4}$$

where $\boldsymbol{K} = (e^{-C(x_i, y_i)})_{i,j=1}^n \in \mathbb{R}^{n \times n}$, and inversion is made pointwise. The Sinkhorn algorithm for OT thus operates in two phases: first, the kernel matrix $\boldsymbol{K}$ is computed, with a cost in $O(n^2 d)$, where $d$ is the dimension of $\mathcal{X}$; then each iteration (4) costs $O(n^2)$. The online Sinkhorn algorithm that we propose mixes these two phases to accelerate convergence (see results in §5.2).

**Consistency and bias.** The OT distance $\mathcal{W}_{C,0}(\alpha,\beta)$ and its regularized version $\mathcal{W}_{C,\varepsilon}(\alpha,\beta)$ can be approximated by the (computable) distance between discrete realizations $\hat{\alpha} = \frac{1}{n}\sum_i \delta_{x_i}$, $\hat{\beta} = \frac{1}{n}\sum_i \delta_{y_i}$, where $(x_i)_i$ and $(y_i)_i$ are i.i.d samples from $\alpha$ and $\beta$. Consistency holds, as $\mathcal{W}(\hat{\alpha}_n, \hat{\beta}_n) \to \mathcal{W}(\alpha,\beta)$. Although this is a reassuring result, the sample complexity of transport in high dimensions with low regularization remains high (see §1).

The estimation of $\mathcal{W}(\alpha,\beta)$ may be improved using several i.i.d sets of samples $(\hat{\alpha}_t)_t$ and $(\hat{\beta}_t)_t$. Those should be of reasonable size to fit in memory and may for example come from a temporal stream. Genevay, Peyré, et al., 2018 use a Monte-Carlo estimate $\hat{\mathcal{W}}(\alpha,\beta) = \frac{1}{T}\sum_{t=1}^T \mathcal{W}(\hat{\alpha}_t, \hat{\beta}_t)$. However, this yields a biased estimation as the distance $\mathcal{W}(\alpha,\beta)$ and the optimal potentials $f^\star = f^\star(\alpha,\beta)$ differ from their expectation under sampling $\mathbb{E}_{\hat{\alpha}\sim\alpha, \hat{\beta}\sim\beta}[\mathcal{W}(\hat{\alpha}, \hat{\beta})]$ and $\mathbb{E}_{\hat{\alpha}\sim\alpha, \hat{\beta}\sim\beta}[f^\star(\hat{\alpha}, \hat{\beta})]$. In contrast, online Sinkhorn consistently estimates the true potential functions (up to a constant) and the Sinkhorn cost.

## 3 OT distances from sample streams

We now introduce an online adaptation of the Sinkhorn algorithm. We construct functional estimators of $f^\star$, $g^\star$ and $\mathcal{W}(\alpha,\beta)$ using successive discrete distributions of samples $(\hat{\alpha}_t)_t$ and $(\hat{\beta}_t)_t$, where $\hat{\alpha}_t \triangleq \frac{1}{n}\sum_{i=n_t+1}^{n_{t+1}} \delta_{x_i}$, with $n_0 \triangleq 0$ and $n_{t+1} \triangleq n_t + n$. The size of the mini-batch $n$ may potentially depends on $t$. $(\hat{\alpha}_t)_t$ and $(\hat{\beta}_t)_t$ may be seen as mini-batches of size $n$ within a training procedure.

### 3.1 Online Sinkhorn iterations

The optimization trajectory $(f_t, g_t)_t$ of the continuous Sinkhorn algorithm given by (3) is untractable as it cannot be represented in memory. The exp-potentials $u_t \triangleq \exp(-f_t)$ and $v_t \triangleq \exp(-g_t)$ are indeed infinitesimal mixtures of kernel functions $\kappa_y(\cdot) \triangleq \exp(-C(\cdot,y))$ and $\kappa_x(\cdot) \triangleq \exp(-C(x,\cdot))$.

We propose to construct finite-memory consistent estimates of $u_t$ and $v_t$ using principles from stochastic approximation (SA) Robbins and Monro, 1951. We cast the regularized OT problem as a root-finding problem of a function-valued operator $\mathcal{F} : \mathcal{C}_+(\mathcal{X}) \times \mathcal{C}_+(\mathcal{X}) \to \mathcal{C}(\mathcal{X}) \times \mathcal{C}(\mathcal{X})$, for which we can obtained unbiased estimates. Optimal potentials are indeed exactly the roots of

$$\mathcal{F} : (u,v) \to \left( u(\cdot) - \int_{y\in\mathcal{X}} \frac{1}{v(y)}\kappa_y(\cdot)\mathrm{d}\beta(y), \quad v(\cdot) - \int_{x\in\mathcal{X}} \frac{1}{u(x)}\kappa_x(\cdot)\mathrm{d}\alpha(x) \right).$$

In particular, the simultaneous Sinkhorn updates rewrites as $(u_{t+1}, v_{t+1}) = (u_t, v_t) - \mathcal{F}(u_t, v_t)$ for all $t$. Importantly, $\mathcal{F}$ can be evaluated without bias using two empirical measures $\hat{\alpha}$ and $\hat{\beta}$, defining

$$\hat{\mathcal{F}}_{\hat{\alpha},\hat{\beta}}(u,v) \triangleq \left( u(\cdot) - \frac{1}{n}\sum_{i=1}^n \frac{1}{v(y_i)}\kappa_{y_i}(\cdot) \quad v(\cdot) - \frac{1}{n}\sum_{i=1}^n \frac{1}{u(x_i)}\kappa_{x_i}(\cdot) \right).$$

By construction, $\mathbb{E}_{\hat{\alpha}\sim\alpha, \hat{\beta}\sim\beta}[\hat{\mathcal{F}}_{\hat{\alpha},\hat{\beta}}] = \mathcal{F}$, and the images of $\hat{\mathcal{F}}$ admit a representation in memory.

**Randomized Sinkhorn.** To make use of a stream of samples $(\hat{\alpha}_t, \hat{\beta}_t)_t$, we may simply replace $\mathcal{F}$ with $\hat{\mathcal{F}}$ in the Sinkhorn updates. This amounts to use noisy soft $C$-transforms in (3), as we set

$$(u_{t+1}, v_{t+1}) \triangleq (u_t, v_t) - \hat{\mathcal{F}}_{\hat{\alpha},\hat{\beta}}(u_t, v_t), \quad \text{i.e.} \tag{5}$$
$$\hat{f}_{t+1} = T_{\hat{\beta}_t}(\hat{g}_t), \qquad \hat{g}_{t+1} = T_{\hat{\alpha}_t}(\hat{f}_{t+1}).$$

$\hat{f}_t$ and $\hat{g}_t$ are defined in memory by $(y_i, \hat{g}_{t-1}(y_i))_i$ and $(x_i, \hat{f}_{t-1}(x_i))_i$. Yet the variance of the updates (5) does not decay through time, hence this *randomized Sinkhorn* algorithm does not converge. However, we show in Proposition 1 that the Markov chain $(\hat{f}_t, \hat{g}_t)_t$ converges towards a stationary distribution that is independent of the potentials $\hat{f}_0$ and $\hat{g}_0$ used for initialization.

**Online Sinkhorn.** To ensure convergence of $\hat{f}_t, \hat{g}_t$ towards some optimal pair of potentials $(f^\star, g^\star)$, one must take more cautious steps, in particular past iterates should not be discarded. We introduce

**Algorithm 1** Online Sinkhorn

> **Input:** Dist. $\alpha$ and $\beta$, learning weights $(\eta_t)_t$, batch sizes $(n(t))_t$ Set $p_i = q_i = 0$ for $i \in (0, n_1]$
> **for** $t = 0, \dots, T-1$ **do**
> Sample $(x_i)_{(n_t, n_{t+1}]} \sim \alpha$, $(y_j)_{(n_t, n_{t+1}]} \sim \beta$.
> Evaluate $(\hat{f}_t(x_i))_{i=(n_t, n_{t+1}]}$, $(\hat{g}_t(y_i))_{i=(n_t, n_{t+1}]}$ using $(q_{i,t}, p_{i,t}, x_i, y_i)_{i=(0,n_t]}$ in (7).
> $q_{(n_t, n_{t+1}], t+1} \leftarrow \log \frac{\eta_t}{n} + (\hat{g}_t(y_i))_{(n_t, n_{t+1}]}, \qquad p_{(n_t, n_{t+1}], t+1} \leftarrow \log \frac{\eta_t}{n} + (\hat{f}_t(x_i))_{(n_t, n_{t+1}]}.$
> $q_{(0,n_t], t+1} \leftarrow q_{(0,n_t], t} + \log(1 - \eta_t), \qquad p_{(0,n_t], t+1} \leftarrow p_{(0,n_t], t} + \log(1 - \eta_t).$
> **Returns:** $\hat{f}_T : (q_{i,T}, y_i)_{(0,n_T]}$ and $\hat{g}_T : (p_{i,T}, x_i)_{(0,n_T]}$

a learning rate $\eta_t$ in Sinkhorn iterations, akin to the Robbins-Monro algorithm for finding roots of vector-valued functions:

$$(\hat{u}_{t+1}, \hat{v}_{t+1}) \triangleq (1 - \eta_t)(\hat{u}_t, \hat{v}_t) - \eta_t \hat{\mathcal{F}}_{\hat{\alpha}_t, \hat{\beta}_t}(\hat{u}_t, \hat{v}_t), \quad \text{i.e.} \tag{6}$$

$$e^{-\hat{f}_{t+1}} = (1 - \eta_t) e^{-\hat{f}_t} + \eta_t e^{-T_{\hat{\beta}_t}(\hat{g}_t)}$$

Each update adds new kernel functions to a non-parametric estimation of $u_t$ and $v_t$. The estimates $\hat{u}_t$ and $\hat{v}_t$ are defined by weights $(p_{i,t}, q_{i,t})_{i \leq n_t}$ and positions $(x_i, y_i)_{i \leq n_t} \subseteq \mathcal{X}^2$:

$$e^{-\hat{f}_t(\cdot)} = \hat{u}_t(\cdot) \triangleq \sum_{i=1}^{n_t} \exp(q_{i,t} - C(\cdot, y_i)), \tag{7}$$

$$e^{-\hat{g}_t(\cdot)} = \hat{v}_t(\cdot) \triangleq \sum_{i=1}^{n_t} \exp(p_{i,t} - C(x_i, \cdot)).$$

The SA updates (6) yields simple vectorized updates for the weights $(p_i, q_i)_i$, leading to Algorithm 1. We perform the updates for $q_i$ and $p_i$ in log-space, for numerical stability reasons.

**Complexity.** Each iteration of online Sinkhorn has complexity $\mathcal{O}(n_t n)$, due to the evaluation of the distances $C(x_i, y_i)$ for all $(x_i)_{(0,n_t]}$ and $(y_i)_{(n_t, n_{t+1}]}$, and the soft $C$-transforms in (7). Online Sinkhorn computes a distance matrix $(C(x_i, y_j))_{i,j \leq n_t}$ on the fly, in parallel to updating $\hat{f}_t$ and $\hat{g}_t$. In total, its computation cost after drawing $n_t$ samples is $\mathcal{O}(n_t^2)$. Its memory cost is $\mathcal{O}(n_t)$; it increases with iterations, which is a requirement for consistent estimation. Randomized Sinkhorn with constant batch-sizes $n$ has a memory cost of $\mathcal{O}(n)$ and a single-iteration computational cost of $\mathcal{O}(n^2)$.

### 3.2 Refinements

**Estimating Sinkhorn distance.** As we will see in §4, the iterations (6) only estimate potential functions up to a constant. This is sufficient for minimizing a loss function involving a Sinkhorn distance (e.g. for model training or barycenter estimation (Staib et al., 2017)), as backpropagating through the Sinkhorn distance relies only on the gradients of the potentials $\nabla_x f^\star(\cdot)$, $\nabla_y g^\star(\cdot)$ (e.g. Cuturi and Peyré, 2018). With extra $\mathcal{O}(n_t^2)$ operations, $(\hat{f}_t, \hat{g}_t)$ may be used to estimate $\mathcal{W}(\alpha, \beta)$ through a final soft $C$-transform:

$$\hat{\mathcal{W}}_t \triangleq \frac{1}{2} \Big( \langle \bar{\alpha}_t, \, f_t + T_{\bar{\alpha}_t}(\hat{g}_t) \rangle + \langle \bar{\beta}_t, \, \hat{g}_t + T_{\bar{\alpha}_t}(f_t) \rangle \Big),$$

where $\bar{\alpha}_t \triangleq \frac{1}{n_t} \sum_{i=1}^{n_t} \delta_{x_i}$ and $\bar{\beta}_t$ are formed of all previously observed samples.

**Fully-corrective scheme.** The potentials $\hat{f}_t$ and $\hat{g}_t$ may be improved by refitting the weights $(p_i)_{(0,n_t]}$, $(q_j)_{(0,n_t]}$ based on all previously seen samples. For this, we update $\hat{f}_{t+1} = T_{\bar{\beta}_t}(g_t)$ and $\hat{g}_{t+1} = T_{\bar{\alpha}_t}(f_t)$. This reweighted scheme (akin to the fully-corrective Frank-Wolfe scheme from Lacoste-Julien and Jaggi, 2015) has a cost of $\mathcal{O}(n_t^2)$ per iteration. It requires to keep in memory (or recompute on-the-fly) the whole distance matrix. Fully-corrective online Sinkhorn enjoys similar convergence properties as regular online Sinkhorn, and permits the use of non-increasing batch-sizes—see §B.1. In practice, it can be used every $k$ iterations, with $k$ increasing with $t$. Combining partial and full updates can accelerate the estimation of Sinkhorn distances (see §5.2).

**Finite samples.** Finally, we note that our algorithm can handle both continuous or discrete distributions. When $\alpha$ and $\beta$ are discrete distributions of size $N$, we can store $p$ and $q$ as fixed-size vectors of size $N$, and update at each iterations a set of coordinates of size $n < N$. The resulting algorithm is a *subsampled* Sinkhorn algorithm for histograms, which is detailed in §B.2, Algorithm 3. We show in §5 that it is useful to accelerate the first phase of the Sinkhorn algorithm.

## 4   Convergence analysis

We show a stationary distribution convergence property for the randomized Sinkhorn algorithm, an approximate convergence property for the online Sinkhorn algorithm with fixed batch-size and an exact convergence result for online Sinkhorn with increasing batch sizes, with asymptotic convergence rates. We make the following classical assumption on the cost regularity and compactness of $\alpha$ and $\beta$.

**Assumption 1.** *The cost $C : \mathcal{X} \times \mathcal{X} \to \mathbb{R}$ is L-Lipschitz, and $\mathcal{X}$ is compact.*

### 4.1   Randomized Sinkhorn

We first state a result concerning the randomized Sinkhorn algorithm (5), proved in §A.2.

**Proposition 1.** *Under Assumption 1, the randomized Sinkhorn algorithm (5) yields a time-homogeneous Markov chain $(\hat{f}_t, \hat{g}_t)_t$ which is $(\hat{\alpha}_s, \hat{\beta}_s)_{s \leqslant t}$ measurable, and converges in law towards a stationary distribution $(f_\infty, g_\infty) \in \mathcal{P}(\mathcal{C}(\mathcal{X})^2)$ independent of the initialization point $(f_0, g_0)$.*

This result follows from Diaconis and Freedman (1999) convergence theorem on iterated random functions which are contracting on average. We use the fact that $T_{\hat{\beta}}(\cdot)$ and $T_{\hat{\alpha}}(\cdot)$ are *uniformly* contracting, independently of the distributions $\hat{\alpha}$ and $\hat{\beta}$, for the variational norm $\| \cdot \|_{\mathrm{var}}$. Using the law of large number for Markov chains (Breiman, 1960), the (tractable) average $\frac{1}{t} \sum_{s=1}^{t} \exp(-\bar{f}_s)$ converges almost surely to $\mathbb{E}[e^{-f_\infty}] \in \mathcal{C}(\mathcal{X})$. This expectation verifies the functional equations

$$\mathbb{E}[e^{-f_\infty}] = \int_y \mathbb{E}[e^{g_\infty(y) - C(\cdot, y)}] \mathrm{d}\beta(y) \quad \mathbb{E}[e^{-g_\infty}] = \int_x \mathbb{E}[e^{f_\infty(x) - C(x, \cdot)}] \mathrm{d}\alpha(x)$$

These equations are close to the Sinkhorn fixed point equations, and get closer as $\varepsilon$ increases, since $\varepsilon\mathbb{E}[\exp(\pm f_\infty/\varepsilon)] \to \mathbb{E}[\pm f_\infty]$ as $\varepsilon \to \infty$. Running the random Sinkhorn algorithm with averaging fails to provide exactly the dual solution, but solves an approximate problem.

### 4.2   Online Sinkhorn

We make the following Robbins and Monro (1951) assumption on the weight sequence. We then state an approximate convergence result for the online Sinkhorn algorithm with fixed batch-size $n(t) = n$.

**Assumption 2.** $(\eta_t)_t$ *is such that $\sum \eta_t = \infty$ and $\sum \eta_t^2 < \infty$, $0 \leqslant \eta_t \leqslant 1$ for all $t > 0$.*

**Proposition 2.** *Under Assumption 1 and 2, the online Sinkhorn algorithm (Algorithm 1) yields a sequence $(f_t, g_t)$ that reaches a ball centered around $f^\star, g^\star$ for the variational norm $\| \cdot \|_{\mathrm{var}}$. Namely, there exists $T > 0$, $A > 0$ such that for all $t > T$, almost surely*

$$\|f_t - f^\star\|_{\mathrm{var}} + \|g_t - g^\star\|_{\mathrm{var}} \leqslant \frac{A}{\sqrt{n}}.$$

The proof is reported in §A.3. It is not possible to ensure the convergence of online Sinkhorn with constant batch-size. This is a fundamental difference with other SA algorithms, e.g. SGD on strongly convex objectives (see Moulines and Bach, 2011). This stems from the fact that the metric for which $\mathrm{Id} - \mathcal{F}$ is contracting is not a Hilbert norm. The constant $A$ depends on $L$, the diameter of $\mathcal{X}$ and the regularity of potentials $f^\star$ and $g^\star$, but not on the dimension. It behaves like $\exp(\frac{1}{\varepsilon})$ when $\varepsilon \to 0$. Fortunately, we can show the almost sure convergence of the online Sinkhorn algorithm with slightly increasing batch-size $n(t)$ (that may grow arbitrarily slowly for $\eta_t = \frac{1}{t}$), as specified in the following assumption.

**Assumption 3.** *For all $t > 0$, $n(t) = \frac{B}{w_t^2} \in \mathbb{N}$ and $0 \leqslant \eta_t \leqslant 1$. $\sum w_t \eta_t < \infty$ and $\sum \eta_t = \infty$.*

**Proposition 3.** *Under Assumption 1 and 3, the online Sinkhorn algorithm converges almost surely:*

$$\|\hat{f}_t - f^\star\|_{\mathrm{var}} + \|\hat{g}_t - g^\star\|_{\mathrm{var}} \to 0.$$

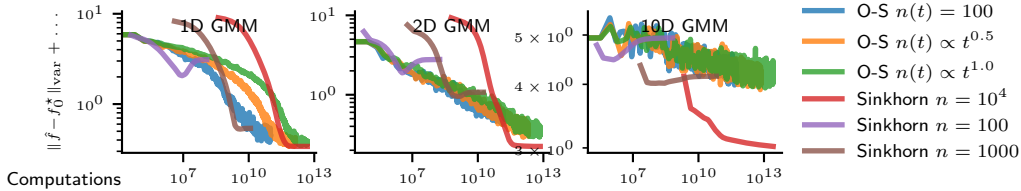

Figure 1: Online Sinkhorn consistently estimate the true regularized OT potentials. Convergence here is measured in term of distance with potentials evaluated on a "test" grid of size $n = 10^4$. Online-Sinkhorn can estimate potentials faster than sampling then scaling the cost matrix.

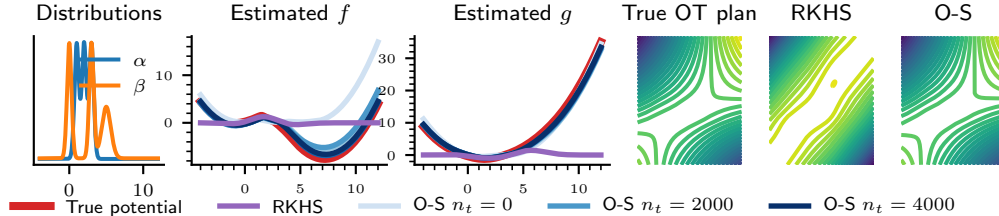

Figure 2: Online Sinhkorn finds the correct potentials over all space, unlike SGD over a RKHS parametrization of the potentials. The plan is therefore correctly estimated everywhere.

The proof is reported in §A.4. It relies on a uniform law of large number for functions (Van der Vaart, 2000, chapter 19) and on the uniform contractivity of soft $C$-transform operator (e.g. Vialard, 2019, Proposition 19). Consistency of the iterates is an original property—Genevay, Cuturi, et al., 2016 only show convergence of the OT value. Finally, using bounds from Moulines and Bach, 2011, we derive asymptotic rates of convergence for online Sinkhorn (see §A.5), with respect to the number of observed samples $N$. We write $\delta_N = \|\hat{f}_{t(N)} - f^\star\|_{\mathrm{var}} + \|\hat{g}_{t(N)} - g^\star\|_{\mathrm{var}}$, where $t(N)$ is the iteration number for which $n_t > N$ samples have been observed.

**Proposition 4.** *For all $\iota \in (0,1)$, $S > 0$ and $B \in \mathbb{N}^\star$, setting $\eta_t = \frac{S}{t^{1-\iota}}$, $n(t) = \lceil Bt^{4\iota} \rceil$, there exists $D > 0$ independant of $N$ and $N_0 > 0$ such that, for all $N > N_0$, $\delta_N \leqslant \frac{D}{N^{\frac{1-\iota}{1+4\iota}}}$.*

Online Sinkhorn thus provides estimators of potentials whose asymptotic sample complexity in variational norm is arbitrarily close to $\mathcal{O}(\frac{1}{N})$. To the best of our knowledge, this is an original property. It also results in a distance estimator $\hat{\mathcal{W}}_N$ whose complexity is arbitrarily close to $\mathcal{O}(\frac{1}{\sqrt{N}})$, recovering existing asymptotic rates from Genevay, Chizat, et al., 2019, for any Lipschitz cost. We derive non-asymptotic rates in §A.5 (see (19)), which make explicit the bias-variance trade-off when choosing the step-sizes and batch-sizes. We also give the explicit form of $D$; it does not depend on the dimension. For low $\varepsilon$, $D$ is proportional to $\exp(\frac{2}{\varepsilon})$; the bound is therefore vacuous for $\varepsilon \to 0$. Note that using growing batch-sizes amounts to increase the budget of a single iteration over time: the overall computational complexity after seeing $N$ samples is always $\mathcal{O}(N^2)$.

## 5 Numerical experiments

The major purpose of online Sinkhorn (OS) is to handle OT between continuous distributions. We first show that it is a valid alternative to applying Sinkhorn on a single realization of continuous distributions, using examples of Gaussian mixtures of varying dimensions. We then illustrate that OS is able to estimate precisely Kantorovich dual potentials, significantly improving the result obtained using SGD with RKHS expansions (Genevay, Cuturi, et al., 2016). Finally, we show that OS is an efficient warmup strategy to accelerate Sinkhorn for discrete problems on several real and synthetic datasets.

### 5.1 Continuous potential estimation with online Sinkhorn

**Data and quantitative evaluation.** We measure the performance of our algorithm in a continuous setting, where $\alpha$ and $\beta$ are parametric distributions (Gaussian mixtures in 1D, 2D and 10D, with

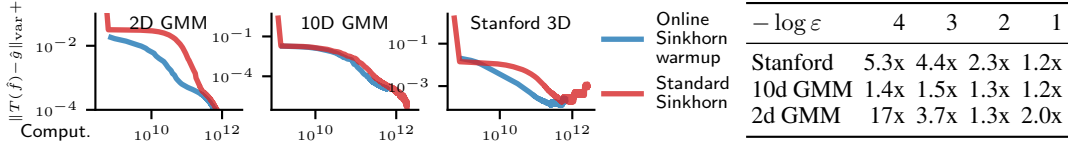

Figure 3: Online Sinkhorn allows to warmup Sinkhorn during the evaluation of the cost matrix, and to speed discrete optimal transport. Table 1: Speed-ups provided by OS vs S to reach a $10^{-3}$ precision.

3, 3 and 5 modes, so that $C_{\max} \sim 1$), from which we draw samples. In the absence of reference potentials $(f^\star, g^\star)$ (which cannot be computed in closed form), we compute "test" potentials $(f_0^\star, g_0^\star)$ on realizations $\hat{\alpha}_0$ and $\hat{\beta}_0$ of size 10000, using Sinkhorn. We then compare OS to Sinkhorn runs of various size , trained on realizations $N = (100, 1000, 10000)$ independent of the reference grid (to avoid reducing the problem to a discrete problem between $\hat{\alpha}_0$ and $\hat{\beta}_0$). To measure convergence, we compute $\delta_t = \|\hat{f}_t - f_0^\star\|_{\mathrm{var}} + \|\hat{g}_t - g_0^\star\|_{\mathrm{var}}$, evaluated on the grid defined by $\hat{\alpha}_0$ and $\hat{\beta}_0$, which constitutes a Monte-Carlo approximation of the error. We evaluate OS with and without full-correction, with different batch-size schedules (see §C.1), as well as the randomized Sinkhorn algorithm. Quantitative results are average over 5 runs. We report quantitative results for $\varepsilon = 10^{-2}$ and non fully-corrective online Sinkhorn in the main text, and all other curves in Supp. Fig. 4. In Supp. Fig. 7, we also report results for OT between Gaussians, which is a simpler and less realistic setup, but for which closed-form expressions of the potentials are known Janati et al., 2020.

**Comparison to SGD.** For qualitative illustration, on the 1D and 2D problem, we consider the main existing competing approach (Genevay, Cuturi, et al., 2016), in which $f_t(\cdot)$ is parametrized as $\sum_{i=1}^{n_t} \alpha_t \kappa(\cdot, x_i)$ (and similarly for $g_t$), where $\kappa$ is a reproducing kernel (typically a Gaussian). This differs significantly from online Sinkhorn, where we express $e^{-f_t}$ as a Gaussian mixture. The dual problem (3) is solved using SGD, with convergence guarantees on the dual energy. As advocated by the authors, we run a grid search over the bandwidth parameter $\sigma$ of the Gaussian kernel to select the best performing runs.

**Earlier potential convergence.** We study convergence curves in Fig. 1, comparing algorithms at equal number of multiplications. OS outperforms or matches Sinkhorn for $N = 100$ and $N = 1000$ on the three problems; it approximately matches the performance of Sinkhorn on $N = 10000$ new iterates on the 1D and 2D problems. On the two low-dimensional problems, online Sinkhorn converges faster than Sinkhorn at the beginning. Indeed, it initiates the computation of the potentials early, while the Sinkhorn algorithm must wait for the cost matrix to be filled. This leads us to study online Sinkhorn as a catalyser of Sinkhorn in the next paragraph. OS convergence is slower (but is still noticeable) for the higher dimensional problem. Fully-corrective OS performs better in this case (see Supp. Fig. 5). We also note that randomized Sinkhorn with batch-size $N$ performs on par with Sinkhorn of size $N$ (Supp. Fig. 6).

**Better-extrapolated potentials.** As illustrated in Fig. 2, in 1D, online Sinkhorn refines the potentials $(\hat{f}_t, \hat{g}_t)_t$ until convergence toward $(f^\star, g^\star)$. Supp. Fig. 8 shows a visualisation for 2D GMM. As the parametrization (7) is adapted to the dual problem, the algorithm quickly identifies the correct shape of the optimal potentials—as predicted by Proposition 3. In particular, OS estimates potentials with much less errors than SGD in a RKHS in areas where the mass of $\alpha$ and $\beta$ is low. This allows to consistently estimate the transport plan, which cannot be achieved using SGD. SGD did not converge for $\varepsilon < 10^{-1}$, while online Sinkhorn remains stable. OS does not require to set a bandwidth.

## 5.2 Accelerating Sinkhorn with online Sinkhorn warmup

The discrete Sinkhorn algorithm requires to compute the full cost matrix $\boldsymbol{C} \triangleq (C(x_i, y_i))_{i,j}$ of size $N \times N$, prior to estimating the potentials $\boldsymbol{f}_1 \in \mathbb{R}^N$ and $\boldsymbol{g}_1 \in \mathbb{R}^N$ by a first $C$-transform. In contrast, online Sinkhorn can progressively compute this matrix while computing first sketches of the potentials. The extra cost of estimating the initial potentials without full-correction is simply $2N^2$, i.e. similar to filling-up $\boldsymbol{C}$. We therefore assess the performance of *online Sinkhorn as Sinkhorn warmup* in a discrete setting. Online Sinkhorn is run with batch-size $n$ during the first iterations, until observing each sample of $[1, N]$, i.e. until the cost matrix $\boldsymbol{C}$ is completely evaluated. From

then, the subsequent potentials are obtained using full Sinkhorn updates. We consider the GMMs of §5.1, as well as a 3D dragon from Stanford 3D scans Turk and Levoy, 1994 and a sphere of size $N = 12000$. We measure convergence using the error $\|T_\alpha(\hat{f}_t) - \hat{g}_t\|_{\text{var}} + \|T_\beta(\hat{g}_t) - \hat{f}_t\|_{\text{var}}$, evaluated on the support of $\alpha$ and $\beta$; this error goes to 0. We use $n(t) = \frac{N}{100}(1 + 0.1t)^{1/2}$—results vary little with the exponent.

**Results.** We report convergence curves for $\varepsilon = 10^{-3}$ in Fig. 3, and speed-ups due to OS in Table 1. Convergence curves for different $\varepsilon$ are reported in Supp. Fig. 9. The proposed scheme provides an improvement upon the standard Sinkhorn algorithm. After $N^2 d$ computations (the cost of estimating the full matrix $C$), both the function value and distance to optimum are lower using OS: the full Sinkhorn updates then relay the online updates, using an accurate initialization of the potentials. The *OS warmed-up* Sinkhorn algorithm then maintains its advantage over the standard Sinkhorn algorithm during the remaining iterations. The speed gain increases as $\varepsilon$ reduces and the OT problem becomes more challenging. Sampling without replacement brings an additional speed-up.

## 6 Conclusion

We have extended the classical Sinkhorn algorithm to cope with streaming samples. The resulting online algorithm computes a non-parametric expansion of the inverse scaling variables using kernel functions. In contrast with previous attempts to compute OT between continuous densities, these kernel expansions fit perfectly the structure of the entropic regularization, which is key to the practical efficiently. We have drawn links between regularized OT and stochastic approximation. This opens promising avenues to study convergence rates of continuous variants of Sinkhorn's iterations. Future work will refine the complexity constants and design adaptive non-parametric potential estimations.

## 7 Acknowledgements

This work was supported by the European Research Council (ERC project NORIA). The work of G. Peyré was supported in part by the French government under management of Agence Nationale de la Recherche as part of the "Investissements d'avenir" program, reference ANR19-P3IA-0001 (PRAIRIE 3IA Institute). A.M thanks Anna Korba, Francis Bach and Gersende Fort for helpful discussions, and Thibault Séjourné for proof-reading and relevant references.

## Broader impact

This work is mostly a theoretical contribution on optimisation for comparing probability distributions. It has therefore no immediate societal impact to be expected.

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
