[Supplementary Material]

# A Proofs

We first introduce two useful known lemmas, and prove the propositions in their order of appearance.

## A.1 Useful lemmas

First, under Assumption 1, we note that the soft $C$-transforms are uniformly contracting on the distribution space $\mathcal{P}(\mathcal{X})$. This is clarified in the following lemma, extracted from Vialard (2019), Proposition 19. We refer the reader to the original references for proofs.

**Lemma 1.** *Unser Assumption 1, let $\kappa = 1 - \exp(-L\operatorname{diam}(\mathcal{X}))$. For all $\hat{\alpha} \in \mathcal{P}(\mathcal{X})$ and $\hat{\beta} \in \mathcal{P}(\mathcal{X})$, for all $f, f', g, g' \in \mathcal{C}(\mathcal{X})$,*

$$\|T_{\hat{\alpha}}(f') - T_{\hat{\alpha}}(f')\|_{\mathrm{var}} \leqslant \kappa\|f - f'\|_{\mathrm{var}}, \quad \|T_{\hat{\beta}}(g) - T_{\hat{\beta}}(g')\|_{\mathrm{var}} \leqslant \kappa\|g - g'\|_{\mathrm{var}}.$$

We will also need a uniform law of large numbers for functions. The following lemma is a consequence of Example 19.7 and Lemma 19.36 of Van der Vaart (2000), and is copied in Lemma B.6 in Mairal (2013).

**Lemma 2.** *Under Assumption 1, let $(f_t)_t$ be an i.i.d sequence in $\mathcal{C}(\mathcal{X})$, such that $\mathbb{E}[f_0] = f \in \mathcal{C}(\mathcal{X})$. Then there exists $A > 0$ such that, for all $n > 0$,*

$$\mathbb{E} \sup_{x \in \mathcal{X}} |\frac{1}{n} \sum_{i=1}^{n} f_i(x) - f(x)| \leqslant \frac{A}{\sqrt{n}}.$$

Finally, we need a result on running averages using the sequence $(\eta_t)_t$. The following result stems from a simple Abel transform of the law of large number, and is established by Mairal (2013), Lemma B.7.

**Lemma 3.** *Let $(\eta_t)_t$ be a sequence of weights meeting Assumption 2. Let $(X_t)_t$ be an i.i.d sequence of real-valued random variables with existing first moment $\mathbb{E}[X_0]$. We consider the sequence $(\bar{X}_t)_t$ defined by $\bar{X}_0 \triangleq X_0$ and*

$$\bar{X}_t \triangleq (1 - \eta_t)\bar{X}_{t-1} + \eta_t X_t.$$

*Then $\bar{X}_t \to_{t \to \infty} \mathbb{E}[X_0]$.*

## A.2 Proof of Proposition 1

*Proof.* We use Theorem 1 from Diaconis and Freedman (1999). For this, we simply note that the space $\mathcal{C}(\mathcal{X}) \times \mathcal{C}(\mathcal{X})$ in which the chain $x_t \triangleq (f_t, g_t)_t$, endowed with the metric $\rho((f_1, g_1), (f_2, g_2)) = \|f_1 - f_2\|_{\mathrm{var}} + \|g_1 - g_2\|_{\mathrm{var}}$, is complete and separable (the countable set of polynomial functions are dense in this space, for example). We consider the operator $A_\theta \triangleq T_{\hat{\beta}}(T_{\hat{\alpha}}(\cdot))$. $\theta \triangleq (\hat{\alpha}, \hat{\beta})$ denotes the random variable that is sampled at each iteration. We have the following recursion:

$$x_{t+2} = A_{\theta_t}(x_t).$$

From Lemma 1, for all $\hat{\alpha} \in \mathcal{P}(\mathcal{X})$, $\hat{\beta} \in \mathcal{P}(\mathcal{X})$, $A_\theta$ with $\theta = (\hat{\alpha}, \hat{\beta})$ is contracting, with module $\kappa_\theta < \kappa < 1$. Therefore

$$\int_\theta \kappa_\theta \mathrm{d}\mu(\theta) < 1, \qquad \int_\theta \log \kappa_\theta \mathrm{d}\mu(\theta) < 0.$$

Finally, we note, for all $f \in \mathcal{C}(\mathcal{X})$

$$\|T_\beta(T_{\hat{\alpha}}(f))\|_\infty \leqslant \|f\|_\infty + 2 \max_{x,y \in \mathcal{X}} C(x, y),$$

therefore $\rho(A_\theta(x_0), x_0) \leqslant 2\|x_0\|_\infty + 2 \max_{x,y \in \mathcal{X}} C(x, y)$ for all $\theta$ $(\hat{\alpha}, \hat{\beta})$. The regularity condition of the theorem are therefore met. Each of the induced Markov chains $(f_{2t}, g_{2t})_t$ and $(f_{2t+1}, g_{2t+1})_t$ has a unique stationary distribution. These stationary distributions are the same: the stationary distribution is independent of the initialisation and both sequences differs only by their initialisation. Therefore $(f_t, g_t)_t$ have a unique stationary distribution $(F_\infty, G_\infty)$. □

### A.3  Proof of Proposition 2

For presentation purpose, we first show that the "slowed-down" online Sinkhorn algorithm converges in the absence of noise. We then turn to prove Proposition 2.

#### A.3.1  Noise-free online Sinkhorn

**Proposition 5.** *We suppose that $\hat{\alpha}_t = \alpha$, $\hat{\beta}_t = \beta$ for all t. Then the updates* (6) *yields a (deterministic) sequence $(f_t, g_t)_t$ such that*

$$\|\hat{f}_t - f^\star\|_{var} + \|\hat{g}_t - g^\star\|_{var} \to 0, \qquad \frac{1}{2}\langle \alpha,\ f_t + T_\alpha(\hat{g}_t)\rangle + \langle \beta,\ \hat{g}_t + T_\beta(f_t)\rangle \to \mathcal{W}(\alpha, \beta).$$

Note that, as we perform *simultaneous* updates, we only obtain the convergence of $f_t \to f^\star + A$, and $g_t \to g^\star$, where $f^\star$ and $g^\star$ are solutions of (1) and $A$ is a constant depending on initialization.

The "slowed-down" Sinkhorn iterations converge toward an optimal potential couple, up to a constant factor: this stems from the fact that we apply contractions in the space $(\mathcal{C}(\mathcal{X}), \|\cdot\|_{var})$ with a contraction factor that decreases sufficiently slowly.

*Proof.* We write $(f_t, g_t)_t$ the sequence of iterates. Given a pair of optimal potentials $(f^\star, g^\star)$, we write $u_t \triangleq f_t - f^\star$, $v_t \triangleq g_t - g^\star$, $u_t^T \triangleq T_\alpha(f_t) - g^\star$ and $v_t^T \triangleq T_\alpha(g_t) - f^\star$. For all $t > 0$, we observe that

$$\begin{aligned}
\max u_{t+1} &= -\log \min \exp(-u_{t+1}) \\
&= -\log \big( \min \big((1 - \eta_t)\exp(-u_t) + \eta_t \exp(-v_t^T)\big)\big) \\
&\leqslant -\log \big((1 - \eta_t)\min \exp(-u_t) + \eta_t \min \exp(-v_t^T)\big) \\
&\leqslant -(1 - \eta_t)\log \min \exp(-u_t) - \eta_t \log \min \exp(-v_t^T) \\
&= (1 - \eta_t)\max u_t + \eta_t \max v_t^T,
\end{aligned}$$

where we have used the algorithm recursion on the second line, $\min f + g \geqslant \min f + \min g$ on the third line and Jensen inequality on the fourth line. Similarly

$$\min u_{t+1} \geqslant (1 - \eta_t)\min u_t + \eta_t \min v_t^T,$$

and mirror inequalities hold for $v_t$. Summing the four inequalities, we obtain

$$\begin{aligned}
e_{t+1} &\triangleq \|u_{t+1}\|_{var} + \|v_{t+1}\|_{var} \\
&= \max u_{t+1} - \min u_{t+1} + \max v_{t+1} - \min v_{t+1} \\
&\leqslant (1 - \eta_t)(\|u_t\|_{var} + \|v_t\|_{var}) + \eta_t(\|u_t^T\|_{var} + \|v_t^T\|_{var}), \\
&\leqslant (1 - \eta_t)(\|u_t\|_{var} + \|v_t\|_{var}) + \eta_t\kappa(\|u_t\|_{var} + \|v_t\|_{var}),
\end{aligned}$$

where we use the contractivity of the soft-$C$-transform, that guarantees that there exists $\kappa < 1$ such that $\|v_t^T\|_{var} \leqslant \kappa\|v_t\|_{var}$ and $\|u_t^T\|_{var} \leqslant \kappa\|u_t\|_{var}$ (Peyré and Cuturi, 2019).

Unrolling the recursion above, we obtain

$$\log e_t = \sum_{s=1}^{t} \log(1 - \eta_t(1 - \kappa)) + \log(e_0) \to -\infty,$$

provided that $\sum \eta_t = \infty$. The proposition follows. $\qquad\square$

*Proof of Proposition 2.* For discrete realizations $\hat{\alpha}$ and $\hat{\beta}$, we define the perturbation terms

$$\varepsilon_{\hat{\beta}}(\cdot) \triangleq f^\star - T_{\hat{\beta}}(g^\star), \qquad \iota_{\hat{\alpha}}(\cdot) \triangleq g^\star - T_{\hat{\alpha}}(f^\star),$$

so that the updates can be rewritten as

$$\begin{aligned}
\exp(-f_{t+1} + f^\star) &= (1 - \eta_t)\exp(-f_t + f^\star) + \eta_t \exp(-T_{\hat{\beta}_t}(g_t) + T_{\hat{\beta}_t}(g^\star) + \varepsilon_{\hat{\beta}_t}) \\
\exp(-g_{t+1} + g^\star) &= (1 - \eta_t)\exp(-g_t + g^\star) + \eta_t \exp(-T_{\hat{\alpha}_t}(f_t) + T_{\hat{\alpha}_t}(f^\star) + \iota_{\hat{\alpha}_t}).
\end{aligned}$$

We denote $u_t \triangleq -f_t + f^\star$, $v_t \triangleq -g_t + g^\star$, $u_t^T \triangleq T_{\hat{\beta}_t}(f_t) - T_{\hat{\beta}_t}(f^\star)$, $v_t^T \triangleq T_{\hat{\beta}_t}(g_t) - T_{\hat{\beta}_t}(g^\star)$.
Reusing the same derivations as in the proof of Proposition 5, we obtain

$$\|u_{t+1}\|_{\text{var}} \leqslant (1 - \eta_t)\|u_t\|_{\text{var}}$$
$$+ \eta_t \log \Big( \max_{x,y\in\mathcal{X}} \exp(\varepsilon_{\hat{\beta}_t}(x) - \varepsilon_{\hat{\beta}_t}(y)) \exp(v_t^T(x) - v_t^T(y)) \Big)$$
$$\leqslant (1 - \eta_t)\|u_t\|_{\text{var}} + \eta_t \|v_t^T\|_{\text{var}} + \eta_t \|\varepsilon_{\hat{\beta}_t}\|_{\text{var}},$$

where we have used $\max_x f(x)g(x) \leqslant \max_x f(x) \max_x f(x)$ on the second line. Therefore, using the contractivity of the soft $C$-transform,

$$e_{t+1} \leqslant (1 - \tilde{\eta}_t)e_t + \frac{\tilde{\eta}_t}{1 - \kappa}(\|\varepsilon_{\hat{\beta}_t}\|_{\text{var}} + \|\iota_{\hat{\alpha}_t}\|_{\text{var}}), \qquad (8)$$

where we set $e_t \triangleq \|u_t\|_{\text{var}} + \|v_t\|_{\text{var}}$, $\tilde{\eta}_t = \eta_t(1 - \kappa)$ and $\kappa$ is set to be the biggest contraction factor over all empirical realizations $\hat{\alpha}_t$, $\hat{\beta}_t$ of the distributions $\alpha$ and $\beta$. It is upper bounded by $1 - e^{-L\text{diam}(\mathcal{X})}$, thanks to Assumption 1 and Lemma 1.

The realizations $\hat{\beta}_t$ and $\hat{\alpha}_t$ are sampled according to the same distribution $\hat{\alpha}$ and $\hat{\beta}$. We define the sequence $r_t$ to be the running average of the variational norm of the (functional) error term:

$$r_{t+1} \triangleq (1 - \tilde{\eta}_t)r_t + \frac{\tilde{\eta}_t}{1 - \kappa}(\|\varepsilon_{\hat{\beta}_t}\|_{\text{var}} + \|\iota_{\hat{\alpha}_t}\|_{\text{var}}).$$

We thus have, for all $t > 0$, $e_t \leqslant r_t$. Using Lemma 3, the sequence $(r_t)_t$ converges towards the scalar expected value

$$r_\infty \triangleq \frac{1}{1 - \kappa}\mathbb{E}_{\hat{\alpha},\hat{\beta}}[\|\varepsilon_{\hat{\beta}}\|_{\text{var}} + \|\iota_{\hat{\alpha}}\|_{\text{var}}] > 0. \qquad (9)$$

We now relate $r_\infty$ to the number of samples $n$ using a uniform law of large number result on parametric functions. We write $\hat{\beta} = \hat{\beta}_n$ to make explicit the dependency of the quantities on the batch size $n$.

Using Lemma 2, we bound the quantity

$$E_n \triangleq \mathbb{E}_{\hat{\beta}_n}\|\varepsilon_{\hat{\beta}_n}\|_{\text{var}} = \mathbb{E}_{\hat{\beta}_n}\|\exp(-T_\beta(g_0^\star)) - \exp(-T_{\hat{\beta}_n}(g_0^\star))\|_\infty$$
$$= \mathbb{E}_{Y_1,\dots Y_n \sim \beta} \sup_{x\in\mathcal{X}} \Big| \frac{1}{n}\sum_{i=1}^n \exp(g^\star(Y_i)) - C(x, Y_i)$$
$$- \mathbb{E}_{Y\sim\beta}[\exp(g_0^\star(Y)) - C(x, Y)] \Big|$$
$$= \mathbb{E} \sup_{x\in\mathcal{X}} |\frac{1}{n}\sum_{i=1}^n \varphi_i(x) - \varphi(x)|,$$

where we have defined $\varphi_i : x \to \exp(g^\star(Y_i) - C(x, Y_i))$ and set $\varphi$ to be the expected value of each $\varphi_i$. The compactness of $\mathcal{X}$ ensures that the functions are square integrable and uniformly bounded. Lemma 2 ensures that there exists $S(g^\star)$ such that

$$E_n \leqslant \frac{S(g^\star)}{\sqrt{n}}.$$

We now bound $\mathbb{E}_{\hat{\beta}_n}\|\varepsilon_{\hat{\beta}_n}\|_{\text{var}}$ using the quantity $E_n$. First, we observe that $\|_{\text{var}} = g_{\text{min}}^\star < g^\star < 0$, and there exists $C_{\text{max}} > 0$ such that $0 \leqslant C(x, y) \leqslant C_{\text{max}}$ for all $x, y \in \mathcal{X}$, thanks to the Assumption 1.

$$\delta \triangleq \exp(-\|g^\star\|_{\text{var}} - C_{\text{max}}) \leqslant \exp(-T_\beta(g^\star)) \leqslant 1$$
$$\exp(-\|g^\star\|_{\text{var}} - C_{\text{max}}) \leqslant \exp(-T_{\hat{\beta}_n}(g^\star)) \leqslant 1,$$

where we have used $g^\star = \|g^\star\|_{\text{var}}$. For all $x \in \mathcal{X}$,

$$|\varepsilon_{\hat{\beta}_n}| = |\log \frac{\exp(-T_{\hat{\beta}_n}(g^\star))}{\exp(-T_\beta(g^\star))}| = \Big| \log \big(1 + \frac{\exp(-T_{\hat{\beta}_n}(g^\star)) - \exp(-T_\beta(g^\star))}{\exp(-T_\beta(g^\star))}\big) \Big|. \qquad (10)$$

We first obtain an upper-bound independent of $n$ with the first equality in (10):

$$||\varepsilon_{\hat{\beta}_n}||_{\text{var}} \leqslant ||\varepsilon_{\hat{\beta}_n}||_{\infty} \leqslant ||g^\star||_{\text{var}} + C_{\max}. \tag{11}$$

We now use the second expression in (10): for $n$ large enough, $E_n < \delta$

$$||\varepsilon_{\hat{\beta}_n}||_{\text{var}} \leqslant \max(\log(1 + \frac{E_n}{\delta}), -\log(1 - \frac{E_n}{\delta})) = -\log(1 - \tilde{E}_n), \tag{12}$$

where we have set $\tilde{E}_n \triangleq \frac{E_n}{\delta}$. On the event $\Omega_n = \{\tilde{E}_n \leqslant \frac{1}{2}\}$, a simple calculation gives $-\log(1 - \tilde{E}_n) \leqslant (2\log 2)\tilde{E}_n \leqslant 2\tilde{E}_n$. Thanks to Markov inequality, $\mathbb{P}[\tilde{E}_n > \frac{1}{2}] \leqslant 2\mathbb{E}[\tilde{E}_n]$. We then split the expectation over the event $\Omega_n$, and use inequalities (12) and (11) on each conditional expectation:

$$\mathbb{E}||\varepsilon_{\hat{\beta}_n}||_{\text{var}} = \mathbb{P}\left[\tilde{E}_n \leqslant \frac{1}{2}\right] \mathbb{E}\left[||\varepsilon_{\hat{\beta}_n}||_{\text{var}}\Big|\tilde{E}_n \leqslant \frac{1}{2}\right] \tag{13}$$

$$+ \mathbb{P}\left[\tilde{E}_n > \frac{1}{2}\right] \mathbb{E}\left[||\varepsilon_{\hat{\beta}_n}||_{\text{var}}\Big|\tilde{E}_n > \frac{1}{2}\right]$$

$$\leqslant \frac{2\varphi(||g^\star||_{\text{var}} + C_{\max})S(g^\star)}{\sqrt{n}}$$

$$\leqslant \frac{4\exp(||g^\star||_{\text{var}} + C_{\max})S(g^\star)}{\sqrt{n}} \triangleq \frac{A(g^\star)}{\sqrt{n}}$$

The constants $S$ depends on the complexity of estimating the functional $x \to \int_y \exp(g^\star(y) - C(x,y))\mathrm{d}\beta(y)$ with samples from $\beta$. A parallel result holds for $\mathbb{E}_{\hat{\alpha}_n}||\iota_{\hat{\alpha}_n}||_{\text{var}}$. Therefore, there exists $A(f^\star), A(g^\star) > 0$ such that $r_\infty \leqslant \frac{A(f^\star) + A(g^\star)}{\sqrt{n}}$. As for all $t > 0$, $e_t \leqslant r_t \to_{t\to\infty} r_\infty$, the proposition follows, writing $A = A(f^\star) + A(g^\star)$.

The constant $A$ is larger than $\exp(C_{\max})$ when $C_{\max} \to \infty$; Hence it behaves at least like $\exp(\frac{1}{\varepsilon})$ when $\varepsilon \to 0$.

Note that we have used twice a corollary of the law of large numbers: once when averaging over $t$ with $t \to \infty$ (Eq. (9)), and once when averaging over $n$ with $n$ finite (Eq. (13)). $\square$

### A.4   Proof of Proposition 3

In the proof of Proposition 2 and in particular Eq. (8), the term that prevents the convergence of $e_t$ is

$$\eta_t(||\varepsilon_{\hat{\beta}_t}||_{\text{var}} + ||\iota_{\hat{\alpha}_t}||_{\text{var}}),$$

which is not summable in general. We can control this term by increasing the size of $\hat{\alpha}_t$ and $\hat{\beta}_t$ with time, at a sufficient rate: this is what Assumption 3 ensures.

*Proof.* From Eq. (8), for all $t > 0$, we have

$$0 \leqslant e_{t+1} \leqslant (1 - \tilde{\eta}_t)e_t + \eta_t(||\varepsilon_{\hat{\beta}_t}||_{\text{var}} + ||\iota_{\hat{\alpha}_t}||_{\text{var}}). \tag{14}$$

Taking the expectation and using the uniform law of large number (13),

$$\mathbb{E}e_{t+1} \leqslant (1 - (1-\kappa)\eta_t)\mathbb{E}e_t + \eta_t\frac{A}{\sqrt{n(t)}} \tag{15}$$

$$= (1 - (1-\kappa)\eta_t)\mathbb{E}e_t + A\eta_t w_t,$$

where we have used the definition of $n(t)$ from Assumption 3 in the last line.

The proof follows from a simple asymptotic analysis of the sequence $(\mathbb{E}e_t)_t$, following recursion (15). For all $t > 0$,

$$\mathbb{E}e_{t+1} - \mathbb{E}e_t = -(1-\kappa)\eta_t\mathbb{E}e_t + A\eta_t w_t \leqslant A\eta_t w_t \tag{16}$$

Therefore, from Assumption 3, $(\mathbb{E}e_{t+1} - \mathbb{E}e_t)_t$ is summable and $\mathbb{E}e_t \to_{t\to\infty} \ell \geqslant 0$. Let's assume $\ell > 0$. Summing (16) over $t$, we obtain

$$\mathbb{E}e_t \leqslant \mathbb{E}e_1 - (1-\kappa)\sum_{s=1}^{t-1}\eta_s\mathbb{E}_s + A\sum_{s=1}^{t-1}\eta_s w_s \to_{t\to\infty} -\infty,$$

which leads to a contradiction. Therefore $\mathbb{E}e_t \to_{t\to\infty} 0$. As $e_t \geqslant 0$ for all $t > 0$, this implies that $e_t \to_{t\to\infty} 0$ almost surely. $\square$

## A.5 Proof of Proposition 4

*Proof.* The proof of Proposition 3 allows us to derive non-asymptotic rates for potential estimations using the online Sinkhorn algorithm. Let us set $\eta_t = \frac{\lambda}{t^a}$, $n(t) = \lceil Bt^{2b} \rceil$ in (14), so that Assumption 3 is met. $\lceil \cdot \rceil$ denotes the ceiling function. We are left to study the recursion (15):

$$\delta_{t+1} \triangleq \mathbb{E}e_{t+1} \leqslant (1 - \frac{\lambda(1-\kappa)}{t^a})\delta_t + \frac{A\lambda}{\sqrt{B}t^{a+b}}$$

Following the derivations of Moulines and Bach (2011, Theorem 2), we have the following bias-variance decomposed upper-bound, provided that $0 \leqslant a < 1$ and $a + b > 1$. For all $t > 0$,

$$\delta_t \leqslant (\delta_0 + \frac{AS}{(a+b-1)\sqrt{B}}) \exp(-\frac{S(1-\kappa)}{2}t^{1-a}) + \frac{2AS}{\sqrt{B}(1-\kappa)t^a}. \tag{17}$$

Let us now relate the iteration number $t$ to the number of seen sample $N$. By definition

$$n_t = \sum_{s=1}^{t} n(s) \leqslant B\sum_{s=1}^{t} s^{2b} + t \leqslant t + \frac{(t+1)^{2b+1}-1}{2b+1} \leqslant (2t)^{2b+1}.$$

Therefore, when we have seen $N$ samples, the iteration number is superior to $t(N)$, and the expected error $\delta_N$ is of the order of $\delta_{t(N)}$, with

$$t(N) = (N/2)^{\frac{1}{2b+1}}. \tag{18}$$

We write $\delta_N = \delta_{t(N)}$. Replacing (18) in (17) yields

$$\delta_n \leqslant (\delta_0 + \frac{A\lambda}{(a+b-1)\sqrt{B}}) \exp\left(-\frac{\lambda(1-\kappa)}{2}(n/2)^{\frac{1-a}{2b+1}}\right) + \frac{2A\lambda}{\sqrt{B}(1-\kappa)(n/2)^{\frac{a}{2b+1}}}. \tag{19}$$

We note that $b$ and $a$ should be as close to $0$ as possible to reduce the bias term, while $a$ should be as close to $1$ and $b$ as close to $0$ as possible to reduce the variance term. Of course, $b$ should remain larger than $1 - a$ to ensure convergence.

To obtain the best asymptotical rates (the error is always dominated by the variance term), we set $a = 1 - \iota$, $b = 2\iota$, with $\iota \gtrsim 0$. This yields

$$\delta_n \leqslant (\delta_0 + \frac{A\lambda}{\iota\sqrt{B}}) \exp\left(-\frac{\lambda(1-\kappa)}{2}(n/2)^{\frac{\iota}{1+4\iota}}\right) + \frac{2A\lambda}{\sqrt{B}(1-\kappa)(n/2)^{\frac{1-\iota}{1+4\iota}}}$$

$$= \mathcal{O}(n^{-\frac{1-\iota}{1+4\iota}}).$$

This rate is as close to the rate $\mathcal{O}(\frac{1}{n})$ as desired. We may then perform a last soft $C$-transform (using the $n_t$ seen samples) over the estimated $f_{t(n)}, g_{t(n)}$ to obtain a estimated solution of the dual optimisation problem (2). The Sinkhorn potentials can therefore be estimated with *fast rates*. Note that the upper bound explodes when $\varepsilon \to 0$, as $C_{\max} \to \infty$, hence $A \to \infty$, and $(1-\kappa) \to 0$. $\qquad \square$

**Estimating the Sinkhorn distance.** The Sinkhorn distance requires to estimate the integral

$$\mathcal{W}(\alpha, \beta) = \int_x f^\star(x)\mathrm{d}\alpha(x) + \int_y g^\star(y)\mathrm{d}\beta(y).$$

At iteration $t(n)$, with empirical realization $\bar{\alpha}_t$ and $\bar{\beta}_t$, containing $n$ samples, we use the estimator

$$\hat{\mathcal{W}}(\alpha, \beta) = \frac{1}{n}\sum_{i=1}^{n} f_{t(n)}(x_i) + \frac{1}{n}\sum_{i=1}^{n} g_{t(n)}(y_i),$$

We can bound the estimation error $|\hat{\mathcal{W}}(\alpha, \beta) - \mathcal{W}(\alpha, \beta)| = \mathcal{O}(\frac{1}{\sqrt{n}})$, dominated by the integral evaluation noise. We thus recover a new estimator of the Sinkhorn distance with the same sample complexity as the batch Sinkhorn estimator (Genevay, Chizat, et al., 2019). Our estimator enjoys an original rate for estimating the potentials in $\|\cdot\|_{\mathrm{var}}$.

---
**Algorithm 2** Fully-corrective online Sinkhorn
---
**Input:** Distribution $\alpha$ and $\beta$, learning weights $(\eta_t)_t$ and batch-sizes $(n(t))_t$. **Set** $p_{i,1} = q_{i,1} = 0$ for $i \in (0, n_1]$

**for** $t = 0, \ldots, T - 1$ **do**
    Sample $(x_i)_{(n_t, n_{t+1}]} \sim \alpha$, $(y_j)_{(n_t, n_{t+1}]} \sim \beta$.
    Evaluate $(\hat{f}_t(x_i))_{i=(0, n_{t+1}]}$, $(\hat{g}_t(y_i))_{i=(0, n_{t+1}]}$ using $(q_{i,t}, p_{i,t}, x_i, y_i)_{i=(0, n_t]}$ in (7).
    $q_{(0, n_{t+1}], t+1} \leftarrow \log \frac{1}{n} + (\hat{g}_t(y_i))_{(0, n_{t+1}]}$,     $p_{(n_t, n_{t+1}], t+1} \leftarrow \log \frac{1}{n} + (\hat{f}_t(x_i))_{(n_t, n_{t+1}]}$.

**Returns:** $\hat{f}_T : (q_{i,T}, y_i)_{(0, n_T]}$ and $\hat{g}_T : (p_{i,T}, x_i)_{(0, n_T]}$
---

---
**Algorithm 3** Online Sinkhorn potentials in the discrete setting
---
**Input:** Distribution $\alpha \in \triangle^N$ and $\beta \in \triangle^N$, $x \in \mathbb{R}^{n \times d}$, $y \in \mathbb{R}^{n \times d}$, learning weights $(\eta_t)_t$

**Set** $p = q = -\infty \in \mathbb{R}^n$.

**for** $t = 1, \ldots, T$ **do**
    $q \leftarrow q + \log(1 - \eta_t)$, $p \leftarrow p + \log(1 - \eta_t)$.
    Sample $J_t \subset [1, N]$, $I_t \subset [1, N]$ of size $n(t)$.
    **for** $i \in J_t$ **do**
        $q_i \leftarrow \log \Big( \exp(q_i) + \exp \big( \log(\eta_t) - \log \frac{1}{N} \sum_{j=1}^{N} \exp(p_j - C(x_j, y_i)) \big) \Big)$.
    **for** $i \in I_t$ **do**
        $p_i \leftarrow \log \Big( \exp(q_i) + \exp \big( \log(\eta_t) - \log \frac{1}{M} \sum_{j=1}^{M} \exp(q_j - C(x_i, y_j)) \big) \Big)$.

Returns $f_T : (q, y)$ and $g_T : (p, x)$
---

## B   Online Sinkhorn variants

### B.1   Fully-corrective scheme

We report the fully-corrective online Sinkhorn algorithm in Algorithm 2. This algorithm also enjoys almost sure convergence, provided that the following assumption is met.

**Assumption 4.** *For all $t > 0$, the total batch-size $n_t = \frac{B}{w_t^2}$ is an integer. The step-size $\eta_t$ and the batch-size $n_t$ grows so that $\sum w_t \eta_t < \infty$ and $\sum \eta_t = \infty$.*

With full correction, the total number of observed samples $n_t$ needs to grow at the same rate as the single-iteration batch-size $n(t)$ in Assumption 3. For $\eta_t = \frac{1}{t^a}$, $a \in (1/2, 1]$, it is sufficient to use a constant batch-size $n(t) = B$ to meet Assumption 4. We then have the following property

**Proposition 6.** *Under Assumption 1 and 4, the fully-corrective online Sinkhorn algorithm converges almost surely:*
$$\|\hat{f}_t - f^\star\|_{\mathrm{var}} + \|\hat{g}_t - g^\star\|_{\mathrm{var}} \to 0.$$

*Proof.* Using the fully-corrective scheme allows to replace $n(t)$ by $n_t = \sum_{s=0}^{t} n(s)$ in (15). The proposition is then obtained in the same way as Proposition 4. $\qquad \square$

### B.2   Online Sinkhorn for discrete distributions

The online Sinkhorn algorithm takes a simpler form with discrete distributions. We derive it in Algorithm 3. We set $\alpha$ and $\beta$ to have size $N$ and $M$, respectively. We evaluate the potentials as

$$g_t(y) = -\log \sum_{j=1}^{N} \exp(p_j - C(x_j, y))$$

$$f_t(x) = -\log \sum_{j=1}^{M} \exp(q_j - C(x, y_j)),$$

Table 2: Schedules of batch-sizes and learning rates that ensures online Sinkhorn convergence.

| Param. schedule | Online Sinkhorn | Fully-corrective online Sinkhorn |
|---|---|---|
| Batch size $n(t) = Bt^b$ | $0 < b$ | $0 \leqslant b$ |
| Step size $\eta_t = \dfrac{1}{t^a}$ | $a \geqslant 1 - \dfrac{b}{2}$ | $\begin{cases} a > \dfrac{1}{2} - \dfrac{b}{2} & \text{and} \quad b < 1 \\ a \geqslant 0 & \text{and} \quad b \geqslant 1 \end{cases}$ |

where $(p_j)_{J \in [1,N]}$ and $(q_j)_{J \in [1,M]}$ are fixed-size vectors. Note that the computations written in Algorithm 3 are in log-space,as they should be implemented to prevent numerical overflows. The sets $|I|$ and $|J|$ can have varying sizes along the algorithm, which allows for example to speed-up the initial Sinkhorn iteration (§5.2). In this case, the cost matrix $\hat{C} = C(x_i, y_j))_{i,j}$ should be progressively recorded along the algorithm iterations.

**B.3   Recapitulation on batch-sizes and learning rates**

To provide practical guidance on choosing rates in batch-sizes $n(t)$ and step-sizes $\eta_t$, we can parametrize $\eta_t = \frac{1}{t^a}$ and $n(t) = Bt^b$ and study what is implied by Assumption 3 and Assumption 4. We summarize the schedules for which convergence is guarantees in Table 2. Note that in practice, it is useful to replace $t$ by $(1 + r\,t)$ in these schedules. We set $r = 0.1$ in all experiments.

Figure 4: Performance of online Sinkhorn for various $\varepsilon$.

## C Extra numerical experiments

We display and describe the supplementary figures mentionned in the main text, as well as experimental details useful for reproduction.

### C.1 Online Sinkhorn and variants

**Grids and details for §5.1.** We set $(\eta_t, n(t)) = \left(\frac{1}{(1+0.1t)^a}, 100(1 + 0.1t)^b\right)$, with $(a, b) = (0, 2)$, $(a, b) = (\frac{1}{2}, 1)$ and $(a, b = 1, 0)$ (constant batch-sizes). Batch Sinkhorn algorithms uses

Figure 5: Performance of fully-corrective online Sinkhorn (O-S) for various $\varepsilon$.

$N = 100, 1000, 10000$. We train Sinkhorn on $t = 5000$ iterations, and train online Sinkhorn long enough to match the number of computations of the large Sinkhorn reference.

**All OS convergence curves.** To complete Fig. 1, Fig. 4 report the performance of online Sinkhorn for $\varepsilon \in \{10^{-4}, 10^{-3}, 10^{-2}, 10^{-1}]\}$. The comparison of performance remains similar to the one produced in the main text.

**Fully-corrective online Sinkhorn.** Fig. 5 reports the performance of fully-corrected online Sinkhorn (FCOS). We observe that the fully-corrective scheme is less noisy than the non-corrected one. It is less efficient than OS on low-dimensional problems, but faster on the 10 dimensional

Figure 6: Performance of randomized Sinkhorn (R-S) for various $\varepsilon$.

problem. For GMM-10D, it outperforms the batch Sinkhorn algorithm with $N = 100, 1000$. Note that we interrupt FCOS for $n_t > 20,000$, as our implementation of the $C$-transform has a quadratic memory cost in $n_t$—this cost can be reduced to a linear cost with more careful implementation [1].

**Randomized Sinkhorn.** Fig. 6 reports the performance of randomized Sinkhorn. In low dimension, randomized Sinkhorn is a reasonable alternative to batch Sinkhorn, as it often outperforms it on average, for the same memory complexity (compare purple to orange curve for instance). In high dimension, batch Sinkhorn tend to perform slightly better.

## C.2    OT between Gaussians

We measure the performance of online Sinkhorn to transport one Gaussian distribution $\alpha$ to another $\beta$. The potentials $f^\star, g^\star$ are known exactly for this problem, which allows to have a strong golden standard. More precisely, adapting the formulae from Janati et al., 2020, assuming $\alpha \sim \mathcal{N}(\mu, A)$ and $\beta \sim \mathcal{N}(\nu, \beta)$ and writing $I$ the identity matrix in $\mathbb{R}^d$, we have

$$C \triangleq (AB + \frac{\varepsilon^2}{4}I)^{1/2}, \quad U \triangleq B(C + \frac{\varepsilon}{2}I)^{-1} - I, \quad V \triangleq A(C + \frac{\varepsilon}{2}I)^{-1} - I$$

$$f^\star : x \to -\frac{1}{2}(x - \mu)^\top U(x - \mu) + x^\top(\mu - \nu)$$

$$g^\star : y \to -\frac{1}{2}(y - \nu)^\top V(y - \nu) + y^\top(\nu - \mu)$$

We compare batch Sinkhorn ($N = 100, 1000, 10000$) to (non fully-corrected) online Sinkhorn, with $n(t) = B$, and $n(t) = B(1 + 0.1t)^{1/2}$, $B = 100$, and $\varepsilon \in \{10^{-4}, 10^{-3}, 10^{-2}, 10^{-1}\}$.

**Results.**    As displayed in Fig. 7, online Sinkhorn outperforms batch Sinkhorn for all tested batch sizes and all $\varepsilon$. It is faster and does not converge towards biased potentials. This suggests that the performance of online Sinkhorn may be underestimated in the previous analyses due to poor potential reference.

## C.3    Illustration of online Sinkhorn potentials on a 2D GMM

The estimate $\hat{f}_t$ is useful to compute the gradient of the Sinkhorn distance $\mathcal{W}(\alpha, \beta)$ with respect to the distribution $\alpha$. This is useful when $\alpha$ is a parametric distribution $\alpha_\theta$, as it allows to compute the gradient of the Sinkhorn distance with respect to $\theta$ using backpropagation. For simplicity, let us assume that $\alpha = \frac{1}{n}\sum_{i=1}^{n}\delta_{x_i}$. Then, for all $i \in [1, n]$,

$$\frac{\partial \mathcal{W}(\alpha, \beta)}{\partial x_i} = \nabla_x\big(x \to f^\star(\alpha, \beta)\big)(x_i),$$

so that $\nabla_x f^\star(\alpha, \beta)$ provides a *displacement field* that can be descended to minimize $\alpha \to \mathcal{W}(\alpha, \beta)$. Such point of view can be extended to general distributions using the mean-field point of view, see e.g. Chizat, 2019; Santambrogio, 2015. Estimating $\nabla_x f^\star(\alpha, \beta)$ is therefore crucial to train e.g. generator networks. Both the online Sinkhorn and the batch Sinkhorn algorithm allow to estimate this vector field, through the plug-in estimator $x \to \nabla_x \hat{f}_t$, easily computed using the form (7) of $\hat{f}_t$.

**Experiment.**    With 2D GMMs, we estimate a reference vector field $\nabla f_0^\star$ using Sinkhorn on $N = 10,000$ samples and qualitatively compare the estimations provided by online Sinkhorn and batch Sinkhorn ($N = 1,000$), for the same number of computations.

**Results.**    We represent the estimations $\nabla_x \hat{f}_t$ in Fig. 8, for $10^8$ computations. We compare them to a reference displacement field, estimated wityh $10^10$ computations. We observe that online Sinkhorn estimates a smoother displacement field than batch Sinkhorn for the same computational budget, that is closer to the reference displacement field. In particular, it is less noisy in low-mass areas. This suggest that online Sinkhorn would be a interesting replacement for batch Sinkhorn in training generative architectures (used by e.g. Genevay, Peyré, et al. (2018)). $\alpha_\theta$ is then defined as the push-forward of some simple measure with a neural network $g_\theta$. We leave this direction for future work.

## C.4    Online Sinkhorn as a warmup process

**Grids and details for §5.2.**    We set $(\eta_t, n(t)) = \big(\frac{1}{(1+0.1t)^a}, 100(1 + 0.1t)^b\big)$, with $(a, b) = (0, 2)$, $(a, b) = (\frac{1}{2}, 1)$ and $(a, b = 1, 0)$ (constant batch-sizes). The batch Sinkhorn algorithm that is used for reference and after warmup uses $N = 10000$. In the reference algorithm, we precompute the distance matrix to save computation. In the warmup algorithm, this distance matrix is filled progressively and then kept in memory to perform $C$-transforms.

Figure 7: Performance of online-Sinkhorn to estimate OT between two Gaussians. Online Sinkhorn systematically outperforms batch Sinkhorn, but in term of speed and correction.

We evaluated OS and fully-corrective OS, and found that fully-corrective was less efficient (due to its higher cost in the early iterations). We evaluated sampling with and without replacement in the warmup phase, and found sampling without replacement to be more efficient.

**All warmup convergence curves.** To complete Fig. 3, we report convergence curves for different $\varepsilon$ in Fig. 9. We find that speed-up increased with $\varepsilon$ and both the 2D and 3D problems, but remains limited for the 10D problem.

Figure 8: Displacement field as defined by the potentials estimated by online-Sinkhorn and Sinkhorn on a 2D GMM. With the same computational budget, online Sinkhorn finds smoother displacement fields than Sinkhorn. Those are closer to the true reference displacement field (we use Sinkhorn on $N = 10000$ to estimate this reference). $\alpha$ and $\beta$ log-likelihood level-lines are displayed in red and blue, while the arrows are proportional to $\nabla_x \hat{f}_t(x) \mathrm{d}\alpha(x)$.

Figure 9: Performance of online-Sinkhorn as warmup for various $\varepsilon$.

## D  Stochastic mirror descent interpretation

The online Sinkhorn can be understood as a stochastic mirror descent algorithm for a non-convex problem. This equivalence is obtained by applying a change of variable in (1), defining

$$\mu \triangleq \alpha \exp(f) \quad \text{and} \quad \nu \triangleq \beta \exp(g). \tag{20}$$

The dual problem (2) rewrites as a minimisation problem over positive measures on $\mathcal{X}$ and $\mathcal{Y}$:

$$- \min_{(\mu,\nu)\in\mathcal{M}^+(\mathcal{X})^2} \mathrm{KL}(\alpha|\mu) + \mathrm{KL}(\beta|\nu) + \langle \mu \otimes \nu, \, e^{-C}\rangle - 1, \tag{21}$$

where the function $\mathrm{KL} : \mathcal{P}(\mathcal{X}) \times \mathcal{M}^+(\mathcal{X}) \triangleq \langle \alpha, \, \log \frac{d\alpha}{d\mu}\rangle$ is the Kullback-Leibler divergence between $\alpha$ and $\mu$. This objective is block convex in $\mu$, $\nu$, but not jointly convex. As we now detail, this problem can be solved using a stochastic mirror descent (Beck and Teboulle, 2003), applied here over the Banach space of Radon measures on $\mathcal{X}$, equipped with the total variation norm.

**Mirror maps and gradient.**  For this, we define the (convex) distance generating function $\mathcal{M}^+(\mathcal{X})^2 \to \mathbb{R}$:

$$\omega(\mu,\nu) \triangleq \mathrm{KL}(\alpha|\mu) + \mathrm{KL}(\beta|\nu).$$

The gradient of this function and of its Fenchel conjugate $\omega^\star : \mathcal{C}(\mathcal{X})^2 \to \mathbb{R}$ yields two *mirror maps*. For all $(\mu,\nu) \in \mathcal{M}^+(\mathcal{X})^2$, $(\varrho,\varphi) \in \mathcal{C}(\mathcal{X})^2$, $\varrho < 0, \varphi < 0$,

$$\nabla\omega(\mu,\nu) = (-\frac{d\alpha}{d\mu}, -\frac{d\beta}{d\nu}) \qquad \nabla\omega^\star(\varrho,\varphi) = (-\frac{\alpha}{\varrho}, -\frac{\beta}{\varphi}).$$

The gradient $\nabla F(\mu, \nu)$ of the objective $F$ appearing in (21) is a continuous function

$$\nabla_\mu F(\mu, \nu) = -\frac{1}{\frac{d\mu}{d\alpha}} + \int_{y \in \mathcal{X}} \frac{d\nu}{d\beta}(y) \exp(-C(\cdot, y)) d\beta(y)$$

and similarly for $\nabla_\nu F$.

**Stochastic mirror descent.** To define stochastic mirror descent iterations, we may replace integration over $\beta$ is by an integration over a sampled measure $\hat{\beta}$. This in turn defines an *unbiased gradient estimate* $\tilde{\nabla} F$ of $\nabla F$, which has bounded second order moments. This absence of bias is crucial to prove convergence of SMD with high probability. Using the mirror maps and the stochastic estimation of the gradient, one has the following equivalence result, whose proofs stems from direct computations.

**Proposition 7.** *The stochastic mirror descent iterations*

$$(\mu_t, \nu_t) = \nabla \omega^\star \left( \nabla \omega(\mu_t, \nu_t) - \eta_t \tilde{\nabla} F(\mu_t, \nu_t) \right)$$

*are equal to the updates* (6) *under the change of variable* (20).

**Interpretation.** It is important to realize that $\mu_t$ and $\nu_t$ do not need to be stored in memory. Instead, their associated potentials $f_t$ and $g_t$ are parametrized as (7). In particular, $\mu_t$ and $\nu_t$ remain absolutely continuous with respect to $\alpha$ and $\beta$ respectively, so that the Kullbach-Leibler divergence terms are always finite. Note that the mirror descent we consider operates in an infinite-dimensional space, as in Hsieh et al. (2018).

Finally, we mention that when computing exact gradients (in the absence of noise) and when using constant step-size of $\eta_t = 1$, the algorithm matches exactly Sinkhorn iterations with simultaneous updates of the dual variables. This provides a novel interpretation on the Sinkhorn algorithm, that differs from the usual Bregman projection (Benamou et al., 2015), and the related understanding of Sinkhorn as a constant step-size mirror descent on the primal objective (Mishchenko, 2019) and on a semi-dual formulation (Léger, 2019).

Note that one can not directly apply the proofs of convergence of mirror descent to our problem, as the lack of convexity of problem (21) prevents their use.

## Footnotes

[1] Using e.g. https://www.kernel-operations.io/keops/index.html