[Reviews · NeurIPS 2020]

Review 1

Summary and Contributions: This paper proposes a framework for online computation of entropy-regularized optimal transport. Compared to current methods that operate on full or mini-batches, this method allows for streaming computation. On the other hand, in contrast with most current methods that operate exclusively on discrete distributions, this one can be used on continuous distributions, for which it relies on a non-parametric representation of the OT dual potentials. The paper proposes various versions of this “online Sinkhorn” scheme, provides convergence results under some assumptions, and presents empirical results on two settings: continuous potential estimation and warm-starting batch Sinkhorn. The results show this is a promising method for online OT computation. --- POST-REBUTTAL -- I thank the authors for their response to my questions/concerns. I do hope they take into account to suggestion to include high-dimensional experiments (even if via embedding), to make the results even more compelling.

Strengths: - Novelty/Originality. Streaming / online computation of optimal transport distances has received much less attention that its batch-computation counterpart, so work which is makes advances in this front is likely to have a strong impact in the community. - Soundness. The proposed method is sound, grounded on solid theory and skillfully leverages ideas from Stochastic Approximation for handling potentials as continuous functions. - Broad/thorough experimental evaluation. Except for the caveat on dimensionality (see weaknesses) The paper does a very good job of validating and comparing the various versions of the proposed approach on synthetic tasks.

Weaknesses: - Motivation. While I think online OT estimation is an interesting challenge from a theoretical standpoint, I feel like the paper is missing a practical/concrete motivation for why/when online OT computation is needed or preferred over batch computation, e.g., perhaps emphasizing the “progressive enrichment” aspect of this framework - No experiments on high-dimensional settings, arguably those for which a streaming setting would be most compelling. Related to the point above, a “real” application that truly benefits from streaming computation of OT would definitely strengthen the paper.

Correctness: The main claims and methods presented in the paper are overall correct. A few smaller issues: - I’m not entirely convinced that using “number of computations” in the x-axis makes sense for Figs 1, 3 etc. Batch-methods that involve some initial pre-computation (e.g., distance matrix computation for batch Sinkhorn), will of course make little to no progress early on. I’m not sure what the the “right” comparison would be, but I don’t think it’s this one - It’s comparing apples to oranges. At any rate, it would be useful to higlight in the plots what part of the computation corresponds to these “no-progress” steps. - In L62 it is claimed that the memory complexity increases lineraly on n_t. Should this be O(n_t^2)? - I could not find a discussion or details on how the learning rate \eta_n is chosen in practice, not sure if I missed it. This is particularly relevant considering Assumption 2.

Clarity: The paper is mostly well written, is relatively easy to follow and understand. Some of the proofs in the Appendix are a bit dry and could use more hand-holding.

Relation to Prior Work: Some related work is missing and/or could be discussed in more detail: * [39] is one of the few recent OT methods for the streaming setting (albeit focused on barycenters). While cited here, it is not described in detailed nor compared against. It would be useful for the reader to have such a comparison in order to better set this work in context. * There is a rich literature on streaming algorithms for the (classic, non-regularized) EMD problem (e.g. [1,2]), which is quite relevant but completely missing from the related work section here. [1] Andoni et al., “Efficient Sketches for Earth-Mover Distance, with Applications“, 2009 [2] Indyk, “Efficient Sketches for Earth-Mover Distance, with Applications”, 2004

Reproducibility: Yes

Additional Feedback: Typos/suggestions: - The thickness of the lines in the the figures makes it hard to compare them. Consider using thinner lines or transparency. - Inconsistent big-O notation in L51.


Review 2

Summary and Contributions: The paper introduces an algorithm for computing optimal transport (OT) distances between two distributions, given access only to streams of samples from the distributions. (this is in contrast to the usual setup, where you take fixed samples from the distributions, then compute OT between those empirical distributions). The authors prove appropriate convergence, sample complexity, and computational complexity results for the algorithm (the latter of which are comparable to Sinkhorn). They also provide compelling experiments showing that the algorithm yields better estimates using the same number of samples, and that it can even effectively warm-start Sinkhorn.

Strengths: The algorithm in this paper seems convincingly better than all alternatives I am aware of for OT between general (e.g. continuous) distributions. The authors back the algorithm with theory that is sensible and reasonably well-explained. The experiments have reasonable breadth and are presented in a clear way (I particularly like the plots in Figure 2 comparing the potentials to those from the baseline from [19]). The authors are also reasonably up-front about the limitations of the algorithm (e.g. that you need to either increase the batch size, or use the fully-corrective scheme in section 3.2). Faster/better OT solvers are highly relevant to NeurIPS, as OT is a tool with wide applicability in ML. As for the significant of this specific algorithm, though it is not likely to displace Sinkhorn e.g. in OT loss functions in deep learning, it is a nice treat that Sinkhorn can be cheaply warm-started via this paper (I imagine this could be used widely).

Weaknesses: There are a few ways in which the experiments could be further improved: Generally, I think more could be done to empirically understand how convergence is affected by various properties of C and the distributions involved. The experiments seem to exclusively show convergence plots in terms of the potentials -- many readers will ask: how does the algorithm here compare in terms of the estimated OT distance itself? I am always a bit suspicious of optimization papers that label the x-axis by iterations (or in this case samples) and not by time, though it may be unfair to compare the algorithm here to highly optimized (e.g. GPU-accelerated) Sinkhorn iterations. Could the authors comment on actual runtime?

Correctness: The theoretical claims seem correct, or at least correct enough that they could be fixed should a bug be found. The experiments also seem correct.

Clarity: For the most part I found the paper a joy to read, though the notation got a bit heavy on page 4. I also got a bit confused at lines 288-289 -- I had trouble figuring out whether it was a two step process (f and g are fit until C is formed as a consequence, then run Sinkhorn) or whether the two steps happened in parallel (f and g are fit, while C is being built on another thread, then when C is done, start running Sinkhorn with whatever f and g are now).

Relation to Prior Work: The paper is well contextualized; I found it clearly demarcated how it is different than prior work.

Reproducibility: Yes

Additional Feedback: The constants in the bounds, e.g. A in proposition 2, look a bit scary -- "It behaves like exp(1/eps)." Any intuition as to whether this can be improved?


Review 3

Summary and Contributions: This paper considers the computation of the entropy regularized optimal transport distance. In particular, unlike the classic Sinkhorn algorithm which assumes an emprical approximation of the underlying measures is available beforehand, this paper proposes an online Sinkhorn method which is able to compute the said distance using a growing sample from the underlying measures. Under the premise that the batch size per round is increasing over time, the proposed method is proved to find the true Sinkhorn potential with a nearly-O(1/n) asymptotic sample complexity.

Strengths: The proposed method has a lower per-iteration computational complexity than the classic Sinkhorn algorithm as it handles the empirical approximation of the underlying measures like streaming data. Concretely, the per-iteration computational complexity is $O(N n_t d)$ where $N$ is the total number of available samples, $d$ is the dimension of the ground space, and $n_t$ is the (growing) batch size in round t. In contrast, under an $O(Nd)$ memory constraint, the classic Sinkhorn algorithm would require $O(N^2d)$ computations per round. Note that the per-iteration cost of the classic Sinkhorn algorithm can be reduced to $O(N^2)$ if the pairwise cost matrix is pre-computed and stored (which of course would entail a $O(N^2 d)$ computation expense and a $O(N^2)$ memory expense). Besides, the author proves that if the size of the sample batch available per iteration grows (but can be arbitrarily slow), the proposed online Sinkhorn algorithm is able to compute the Sinkhorn potential at the rate of $O(1/N)$ where N is the number of the total available samples.

Weaknesses: I find the Proposition 4 surprising. In the limiting case $\iota$ -> 0, P4 states that the sample complexity is $O(1/N)$, but P2 means that for $\iota=0$, the proposed online Sinkhorn algorithm only converges to a neighbor of the true Sinkhorn potential. So there is a bit jump here. ======update after author response======= In Prop 4., if we take \iota to be arbitrarily close to 0, the proposed algorithm has the best performance O(1/N). However, if we take \iota to be exactly zero, according to prop 2, there is no convergence guarantee. So what should we expect in practice? When taking a sufficiently small \iota, all finite iterations have the same batch size, should it converge or not? Either way, one of the proposition is violated. The authors should address this in their revision.

Correctness: I am not sure about that. Please see the discussion in the Weaknesses section.

Clarity: I do not see how Proposition 2 agrees with the limiting case of Proposition 4. Maybe the author can elaborate on that.

Relation to Prior Work: Yes

Reproducibility: Yes

Additional Feedback:


Review 4

Summary and Contributions: The authors propose a novel scheme for online computation of the sinkhorn divergence, of great interest for the ML community. Claims (different modes of convergence of the algorithm) are substantiated by thorough theoretical analysis and experiments on simple datasets of up to 10d. Post author feedback Authors have addressed my points but I won't update score since it is already very good.

Strengths: The paper has many strengths and is bound to generate impact in the machine learning community. Authors thoroughly elaborate the underlying theory and make the distinction (through a number of convergence results) of random and online sinkhorn. Specifically, authors appeal to stochastic optimization to 'solve' the lack of almost sure convergence of the 'naive' randomized algorithm. There is an is in depth discussion about the trade-offs entailed by the fact that the batch size has to grow to obtain convergence. Authors also show an interesting connection with mirror descent. This is a timely contribution, this paper addresses a relevant computational issue with Sinkhorn divergences. Comparisons with existing methods are sensible and experiments insightful (but see below). A number of computational 'tricks' (as warm starts) are also presented and the impactful is measured. This adds value. The supplement is comprehensive and contains very nice additional experiments.

Weaknesses: Experiments are performed in only simple cases. I understand this is mostly a theory paper but for the sake of impact the paper would really benefit from including experiments with more complex dataset, where the theoretical gains are observed in practice.

Correctness: As far as I can judge there are no mistakes

Clarity: Yes. Perhaps the notation is a bit hard to understand with n_t and \eta_t Line 170 is a bit confusing. Could you clarify what you mean by "Soft C transform"

Relation to Prior Work: The paper cites most of exciting work but it misses a citation to relevant similar work, though from a more statistic perspective https://arxiv.org/pdf/1812.09150.pdf

Reproducibility: Yes

Additional Feedback: Authors may discuss on the relation between their work and the one from Bercu and Bigot (see above). Also, does your intuition about stochastic EM relates somehow to the recently develop Sinkhorn EM algorithm? https://arxiv.org/abs/2006.16548 Authors may also cite https://arxiv.org/abs/1905.11882 as this works is stated for general spaces. The above paper contains a more general sample complexiy statement. In Figure 1, authors may provide entire traces for all methods.

[Author Response · NeurIPS 2020]

We thank the reviewers for their insightful comments. We address their interrogations and comments below.

**R1.** - *Practical/concrete motivation for why/when online OT computation is needed.* Online OT estimation is useful for
training generative models. In this case, the samples are renewed at each iteration of a training algorithm, and may be
used to better evaluate the distance to minimize. Any applications that requires to estimate OT distances between large
point clouds can benefit from online OT estimation, that accelerates training. We will better motivate our work.

- *No experiments on high-dimensional settings, arguably those for which a streaming setting would be most compelling.*
This is indeed a weakness in our experiments, as we have measured performance up to $d = 10$. Real-world ML
applications would typically consider points in latent spaces, with typical dimension $d = 128$, with a certain "manifold"
structure that makes OT estimation possible. We may consider the output of a trained CNN on CIFAR10 images.

- *In L62 it is claimed that the memory complexity increases lineraly on $n_t$. Should this be $O(n_t^2)$?* The memory
complexity is linear in $n_t$, as each potential is represented in memory by $n_t$ points and weights. We will clarify.

- *I'm not entirely convinced that using "number of computations" in the x-axis makes sense for Figs 1, 3 etc.* We measure
the number of computations needed to obtain a *first estimate* of the OT potentials, which is roughtly proportional to
wall-clock time (see answer to **R2**). It is of course higher for batch method than online method. Our intent in Fig. 3 is
to show that online Sinhkorn efficiently warms up OT computation. We will clarify.

- *I could not find a discussion or details on how the learning rate $\eta_n$ is chosen in practice.* We give practical
recommendation regarding step-sizes and batch-sizes in Appendix B.3, and in particular Table 1. In experiments, we
found that setting $n(t) \propto (1 + 0.1t)^{1/2}$, and $\eta_t = 1$ work best, although the range of usable exponents is rather wide.
We will present Table 1 in the main text for clarification. See also App. C for details on hyper-parameters.

- *Further references.* We thank the reviewer for his insightful refrences on streaming method for EMD estimation, that
we will discuss in the related work section. In the batch or online setting, regularization permits a faster estimation of
OT distances, relying only on matrix-vector products. [39] fixes a spatial grid for the estimated barycenter, unrelated to
observed samples, while we define potentials based on observed samples. We will discuss this in details.

**R2.** - *How is convergence affected by [...] the distributions involved.* We have tried to give more insight on this aspect
in Appendix C. As predicted in the analysis, online Sinkhorn converges more slowly for lower $\epsilon$ (or equivalently, less
regular $C$, Fig. 5). For Gaussians distributions, online Sinkhorn outperforms batch Sinkhorn in all cases (Fig. 7).

- *Could the authors comment on actual runtime?* With proper GPU implementation of online Sinkhorn (using the
*pyKeops* library), the C-transform wall-clock time is indeed roughly in $O(n(t)^2)$. We have compared online Sinkhorn
to batch Sinkhorn in term of wall-clock time, and found similar curves as reported in the paper, using batch-sizes larger
than 1000. Batch Sinkhorn remains faster for small problems ($N < 10^4$), for which $C$ can be precomputed and held in
GPU memory. We will add wall-clock time experiments to the appendix.

- *Confusion l.288-289.* $f$ and $g$ are fit until $C$ is formed, and we then run batch Sinkhorn. We will clarify.

- *"It behaves like exp(1/eps)." Any intuition as to whether this can be improved?* This is a difficult open question.
Entropic regularization improves the sample complexity of Optimal Transport, going from a rate in $O(n^{-1/d})$ to a rate
in $O(1/\sqrt{n})$. This improvement is not free: as noted in [18], the constants before these rates explodes as $\exp(1/\epsilon)$ as
$\epsilon \to 0$. Both [18] and our work relies on the contractance modulus of the soft C-transform, hence the similar conclusion.

**R3.** - *The per-iteration cost of the classic Sinkhorn algorithm can be reduced [...].* We have been too elusive on this
aspect. Online Sinkhorn proves most useful in the case where the pairwise cost matrix must be computed on the fly due
to memory constraints (and serves as a sound warmup otherwise). We will recall and discuss this observation.

- *I find the Prop. 4 surprising.* In Prop. 4, we assume that $\iota > 0$, and therefore that the batch-size goes to infinity. This
is sufficient to ensure convergence, as the variance terms introduced by sampling are summable. For fixed batch-sizes,
convergence cannot be guaranteed, due to the fact that $\sum_t \frac{1}{t}$ is not summable. The proof of Prop. 4 established a
classical recursion between error terms, and requires $\iota > 0$ to conclude. We will discuss Prop. 4 more thoroughly.

**R4.** - *Experiments are performed in only simple cases.* This is a limitation of our work. The lack of gold-standard
for estimating continuous OT distances makes it hard to evaluate our method on hard settings, as we are forced to
approximate this gold-standard with very long runs of Sinkhorn algorithm. See also answer to **R1**.

- *Soft C-transform.* This term refers to Eq. (3). We will clarify.

- *Related work.* Bercu and Bigot's work is indeed relevant. It tackles the simpler problem of semi-discrete OT, that
rewrites as a expected risk-minimization problem. A single finite dimensional potential must be estimated, which
can be done through gradient descent. We will refer to Mena and Weed's refined sample complexities. The E-step of
Sinkhorn-EM could be implemented using online Sinkhorn, with potential gain from warm-starting.

[Meta-Review · NeurIPS 2020]

This paper gives a streaming estimator of entropy-regularized optimal transport, which allows for continuous distributions, gives better estimates of the transport plan than previous approaches, and enjoys strong theoretical guarantees. Although the numerical experiments are somewhat limited, they are more than sufficient to show the value of the new estimator. As a whole, then, the paper is clearly worthy of inclusion at NeurIPS, and should be quite useful to practitioners in certain settings. One remaining comment was brought up in discussion: Proposition 2 and Proposition 4, though not contradictory, seem to give different conclusions about a constant batch size with \iota nearly 0: Proposition 2, constant batch size with \iota = 0, gives no convergence, while taking \iota \to 0 in Proposition 4 gives n(t) -> B, a constant batch size, and a convergence rate of D / N. It would be helpful to add a brief discussion explaining this unintuitive discontinuity in the final version of the paper. (Is it simply that D and/or N_0 explode as iota -> 0?) Additionally, one very minor note: in your author response you used "his" to refer to a reviewer; since you presumably do not know the gender of the reviewer, it is preferable to use gender-neutral language, the preferred form of which in contemporary English is "they"/"them"/"their".